# Exploring the Potential Molecular Mechanisms of Interactions between a Probiotic Consortium and Its Coral Host

Phillipe M. Rosado,[a] Pedro M. Cardoso,[a] João G. Rosado,[a] Júnia Schultz,[a,b] Ulisses Nunes da Rocha,[c] Tina Keller-Costa,[d,e] Raquel S. Peixoto[a]

[a]Red Sea Research Centre, King Abdullah University of Science and Technology, Thuwal, Saudi Arabia
[b]Biological and Environmental Science and Engineering Division, King Abdullah University of Science and Technology, Thuwal, Saudi Arabia
[c]Department of Environmental Microbiology, Helmholtz Centre for Environmental Research-UFZ, Leipzig, Germany
[d]Institute for Bioengineering and Biosciences, Instituto Superior Técnico, Lisbon, Portugal
[e]Institute for Health and Bioeconomy, Instituto Superior Técnico, Lisbon, Portugal

**ABSTRACT** Beneficial microorganisms for corals (BMCs) have been demonstrated to be effective probiotics to alleviate bleaching and mitigate coral mortality *in vivo*. The selection of putative BMCs is traditionally performed manually, using an array of biochemical and molecular tests for putative BMC traits. We present a comprehensive genetic survey of BMC traits using a genome-based framework for the identification of alternative mechanisms that can be used for future *in silico* selection of BMC strains. We identify exclusive BMC traits associated with specific strains and propose new BMC mechanisms, such as the synthesis of glycine betaine and ectoines. Our roadmap facilitates the selection of BMC strains while increasing the array of genetic targets that can be included in the selection of putative BMC strains to be tested as coral probiotics.

**IMPORTANCE** Probiotics are currently the main hope as a potential medicine for corals, organisms that are considered the marine "canaries of the coal mine" and that are threatened with extinction. Our experiments have proved the concept that probiotics mitigate coral bleaching and can also prevent coral mortality. Here, we present a comprehensive genetic survey of probiotic traits using a genome-based framework. The main outcomes are a roadmap that facilitates the selection of coral probiotic strains while increasing the array of mechanisms that can be included in the selection of coral probiotics.

**KEYWORDS** beneficial microorganisms for corals (BMCs), coral probiotics, genomes, mechanisms, symbiosis, molecular interactions, holobiont, BMCs, probiotics

Changes in microbial diversity and activity affect the resilience of their hosts and therefore their ability to respond to climate change (1). Corals are model examples because associated microorganisms play a critical role in their development and growth (2, 3), in the control of pathogens (4–7), and in biogeochemical cycles (8) such as nitrogen, carbon, and sulfur cycling (9, 10), also providing access to different types of nutrients (11).

It is well known that climate change can destabilize host-associated microbiomes, leading to a state of dysbiosis, which is a disruption of the symbiotic relationships between the host and its associated microorganisms (12, 13). However, this cause-and-effect relationship is still poorly explored. Some coral species are tolerant to environmental changes and are able to maintain a relatively stable microbiome even at low pH (14) or elevated temperatures (15, 16), while other species go through a microbiome restructuring in response to impacts, such as increased seawater temperature (17, 18) and exposure to wastewater (19) and oil spills (20, 21). Some coral microbiomes can return to their original

Address correspondence to Raquel S. Peixoto, raquel.peixoto@kaust.edu.sa.

The authors declare no conflict of interest.

microbiome state after the stressor is removed (22, 23), while others change irreversibly to a new microbiome state that can be either beneficial or harmful to the holobiont (24).

The extensive taxonomic and metabolic diversity of the microbiome, as well as its plasticity and notably shorter generation times (compared to the coral host), provides a possible role for microorganisms in the adaptive response of the holobiont (25–30). Advantageous modifications to the microbiome could potentially be transferred vertically to subsequent generations of corals, thereby increasing the resistance of future generations to environmental stresses (31–34).

A growing appreciation of the role that microorganisms play in maintaining animal health and ecosystem functioning (27), along with the realization of possible implications for the conservation and management of endangered species (28, 35), has led some researchers to explore whether microbiome manipulation could be used to help protect and restore coral reefs (3, 6, 7, 10, 20, 21, 26, 36–39). One approach is the use of specific probiotics for corals (40), the so-called beneficial microorganisms for corals (BMCs) (26). Different members of the coral microbiome may perform the same beneficial functions (i.e., functional redundancy) (10). Therefore, one of the aims of using BMCs is to increase (or retain, as some beneficial microbes may be replaced by pathogens, during thermal stress events) the number of common, abundant, and native microorganisms that can contribute certain functions in corals undergoing stress (10, 26, 27).

Rosado et al. (7) tested the application of a BMC consortium composed of seven bacteria, including five belonging to the *Pseudoalteromonas* genus, one to the *Cobetia* genus, and one to the *Halomonas* genus (Table S1). All of them were isolated from the coral *Pocillopora damicornis* and used in a mesocosm experiment, mimicking the increase in temperature and the addition of a known coral pathogen (*Vibrio coralliilyticus* strain ATCC BAA-450). The experiment was conducted at two different temperatures (26°C and 30°C), comparing four treatments (BMC, BMC with a pathogen, pathogen, and control). The results of their study showed that a BMC consortium could indeed minimize the effects of both coral bleaching and pathogen challenge. Metrics used to calculate bleaching severity were significantly reduced in corals inoculated with a BMC consortium in contrast to the control treatments or treatments with pathogen addition without BMC consortia, which exhibited strong bleaching signals. For instance, the *Fv/Fm* ratios, a metric that assesses the Symbiodiniaceae photosystem function, were higher in treatments with inoculation of the probiotic consortium compared to the control and the treatment with pathogen inoculation.

The purpose of this work is to perform an *in silico* analysis of the genomes of the *P. damicornis* BMCs used by Rosado et al. (7), aiming to identify potential molecular mechanisms of interaction between members of the consortium and the host that may in turn guide further selections of novel BMCs. In this regard, our *in silico* analysis can support the development of a framework for the selection of customized consortia with specific BMC characteristics for specific hosts and stress conditions, which can accelerate and optimize the selection of BMC consortia. However, it is important to emphasize that the beneficial characteristics for corals analyzed in this work are mainly theoretical and based on the literature and therefore require validation by, for example, combined physiological and molecular (such as metatranscriptomics and other omics) monitoring during laboratory experiments and field trials.

## RESULTS

**Genomic features of the BMC strains.** A combined total of 11,789,856 clean and trimmed high-quality reads were obtained from Illumina sequence reads. The genome assembly of the seven BMC strains performed using SPAdes resulted in a total of 68 to 222 contigs/strain (total contigs obtained). CheckM analysis revealed a high genome completeness (>99% and 100% complete) and a low level of contamination (<1%) for all genomes, except BMC 1, which, after using RefineM to improve the quality, reached a completeness of 87.58% while contamination dropped from 31.25% to 0.19%.

General features from the draft genomes of the BMC strains are summarized in Table S4.

The *Pseudoalteromonas* sp. genomes of BMC strains 1 to 5 presented very similar characteristics, with a G + C content of 41% and an average size of 4.5 Mbp. Notably, the values of the number of contigs, tRNAs, rRNAs, and coding genes were lower in BMC 1 compared to BMCs 2, 3, 4, and 5, probably due to the refinement of this genome, which resulted in a loss of 12% of its gene content. The genomes of BMC strains 6 and 7, previously identified as different genera (*Cobetia* and *Halomonas*, respectively), showed very similar characteristics to each other, such as a G + C content of 62.47% and 61.60%, respectively, and an average genome size of 3.9 Mbp.

**Taxonomic attribution.** To identify each BMC strain, two different genome alignment methods (FastANI and Genome-to-Genome Distance Calculator [GGDC]) were used to mimic conventional DNA-DNA hybridization (DDH). Pairwise comparison between BMCs 1 to 5 using the GGDC index obtained a similarity value of 100%, while the average nucleotide identity (ANI) indicated a similarity between 97.28% and 99.99%, confirming that they belong to the same species (Table S2).

Four publicly available *Pseudoalteromonas* genomes obtained a DDH estimate higher than 70% (*Pseudoalteromonas* sp. CO109Y, *Pseudoalteromonas shioyasakiensis* JCM 18891, *P. shioyasakiensis* M1400201, and *Pseudoalteromonas* sp. P1-8), and five obtained an ANI value higher than 95% (the previous four *Pseudoalteromonas* plus *P. shioyasakiensis* SDCH90) when aligned with all *Pseudoalteromonas* BMCs (Table S2). The cutoff values used for organisms to be considered the same species were 70% and 95% for the DDH estimate and ANI value, respectively. Based on these results, it is possible to conclude that BMCs 1 to 5 belong to the same species, *P. shioyasakiensis*.

The same analyses were performed for BMCs 6 and 7 (Table S2), previously identified (16S rRNA gene analysis) as *Cobetia marina* and *Halomonas taeanensis*, respectively (7). Ten genomes belonging to the genus *Cobetia* were used in the comparison with the genome of BMC 6 (Fig. 1; Table S2). The ANI taxonomic assignment method indicated that all 10 genomes shared more than 95% similarity. However, the DDH estimate indicated that none of the 10 genomes showed more than 70% similarity. The similarity values obtained from the comparison between BMC 7 and publicly available reference/closest *Halomonas* genomes were below the cutoff value for species identification for both ANI and GGDC methods (i.e., 91.31% and 41.6%, respectively) (Fig. 2; Table S2).

**Phylogenomic tree assembly using the draft genomes of BMCs.** The phylogenomic relationships between the seven BMC genomes and the closest publicly available genomes were evaluated. In the assembly of the phylogenomic tree of *Pseudoalteromonas* sp. BMCs, 90 genomes belonging to the genus *Pseudoalteromonas* were used (Fig. S1). In this phylogenomic tree, a well supported clade, formed by the BMCs 1, 2, 3, 4, and 5 and 17 other *Pseudoalteromonas* sp. strains, was identified. Subsequently, another tree was assembled using only the genomes of this *P. shioyasakiensis* BMC clade (Fig. 3). Of the 17 strains used in this phylogenomic tree, 10 were isolated from a host, while 7 were isolated from either sediment or seawater. Among the hosts were corals, cnidarians, mollusks, sponges, and echinoderms.

For the phylogenomic inference of *Cobetia* sp. BMC 6, another 24 genomes of strains belonging to the genus *Cobetia* were used. *Cobetia* sp. BMC 6 formed a clade with 10 other *Cobetia* sp. genomes. Of these 10 strains, only 2 were isolated from a host (seagrass and mollusk), while 4 were isolated from sediment, and 4 were isolated from seawater (Fig. 1).

In the *Halomonas* sp. BMC tree, 39 other publicly available genomes were included. The formation of a small clade between *Halomonas* sp. BMC 7 and three other strains of *Halomonas* sp. (YLGW01, *H. taeanensis* USBA-857, and *H. taeanensis* BH539) was observed (Fig. 2). None of the three strains were isolated from a host; one was isolated from sediment, and the other two, USBA-857 and BH539, were from seawater.

**Gene functions with potential benefits for corals.** Genes encoding proteins that may be related to potentially beneficial characteristics for corals (26, 41) or for their endosymbiont algae (42) were selected using PATRIC. Subsystems related to overall

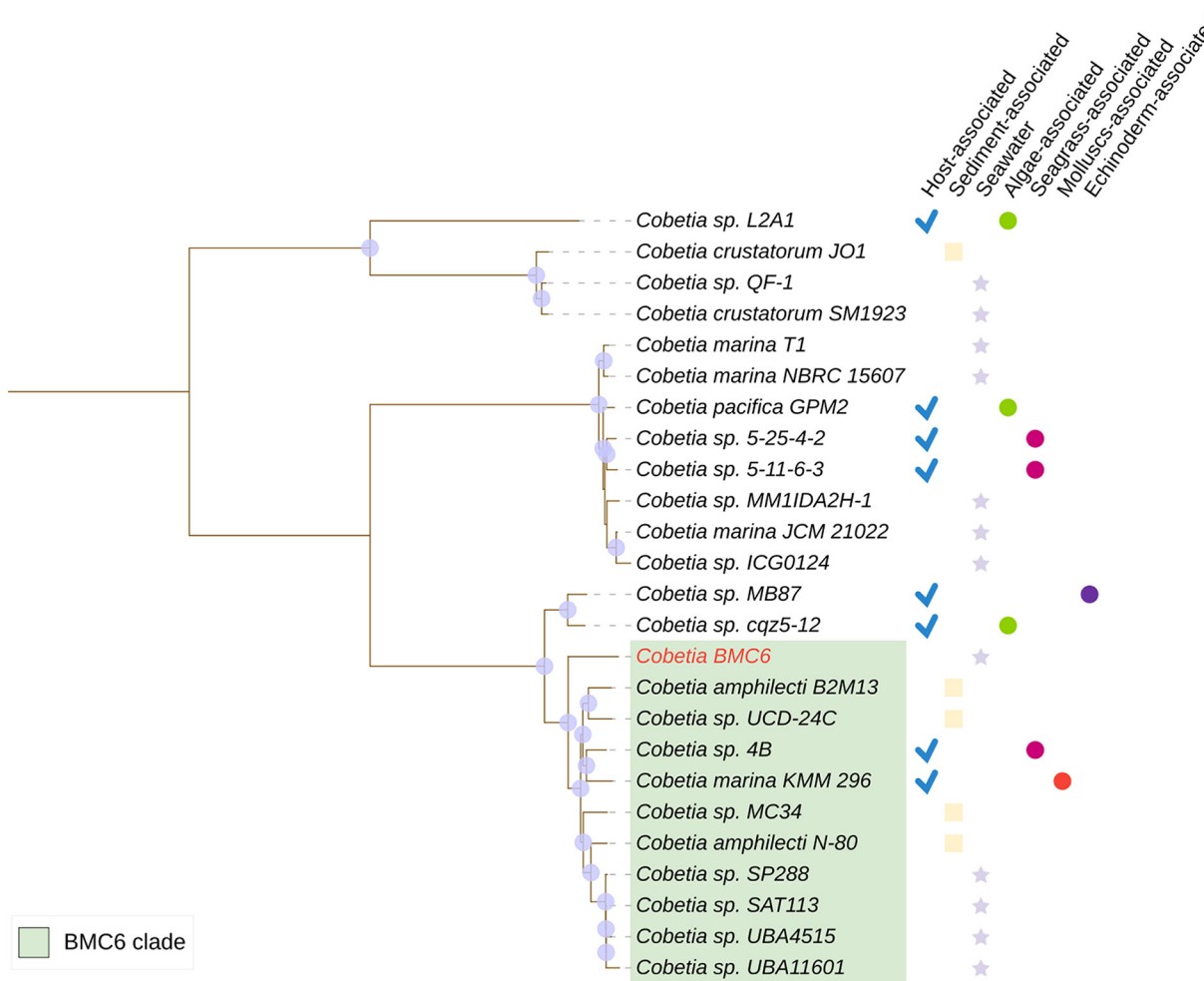

**FIG 1** Phylogenomic inference of publicly available genomes from 24 *Cobetia* strains and the genomes of beneficial microorganisms for coral strain 6 (BMC6) (in red), totaling 25 genomes. The green-shaded area represents the strains that form a monophyletic clade with BMC6. The tree was assembled from the comparison of 1,000 proteins through the codon tree method of the PATRIC platform that selects global protein families (PGFams) as homology groups and analyzes aligned proteins and DNA encoding single-copy genes using the RAxML program. The best protein model found by RAxML to build this tree was Jones-Taylor-Thornton (JTT). Purple dots on the branch length correspond to the bootstrap value of 100. The icons on the right side of the strains represent their isolation sources.

potentially beneficial traits described by Peixoto et al. (26, 41) and Matthews et al. (42) were screened; genes linked to oxidative stress, nitrogen cycling, cobalamin (vitamin B₁₂) synthesis, siderophore production (i.e., production of iron-chelating compounds), dimethylsulfoniopropionate (DMSP) degradation, glycine betaine production, and ectoine synthesis were identified (Table S5).

The subsystems of interest found in the genomes of *P. shioyasakiensis* BMCs 1 to 5 included oxidative stress protection, cobalamin synthesis, and glycine betaine synthesis. The number of annotated genes belonging to these subsystems was similar among all BMC bacteria of the species. *Cobetia* sp. BMC 6 and *Halomonas* sp. BMC 7 shared subsystems related to oxidative stress, nitrogen cycling, siderophore synthesis, glycine betaine production, ectoine synthesis, and DMSP degradation. *Halomonas* sp. BMC 7 also harbored genes for cobalamin synthesis, which were absent in *Cobetia* sp. BMC 6.

**Pangenome comparison and categorization of singleton genes.** A pangenomic comparison was performed to identify genes that were unique to specific BMC strains. Pangenomes were independently generated for each BMC clade (Fig. 4 to 6). Seventeen additional *Pseudoalteromonas* strains were added to the pangenome of *P. shioyasakiensis*

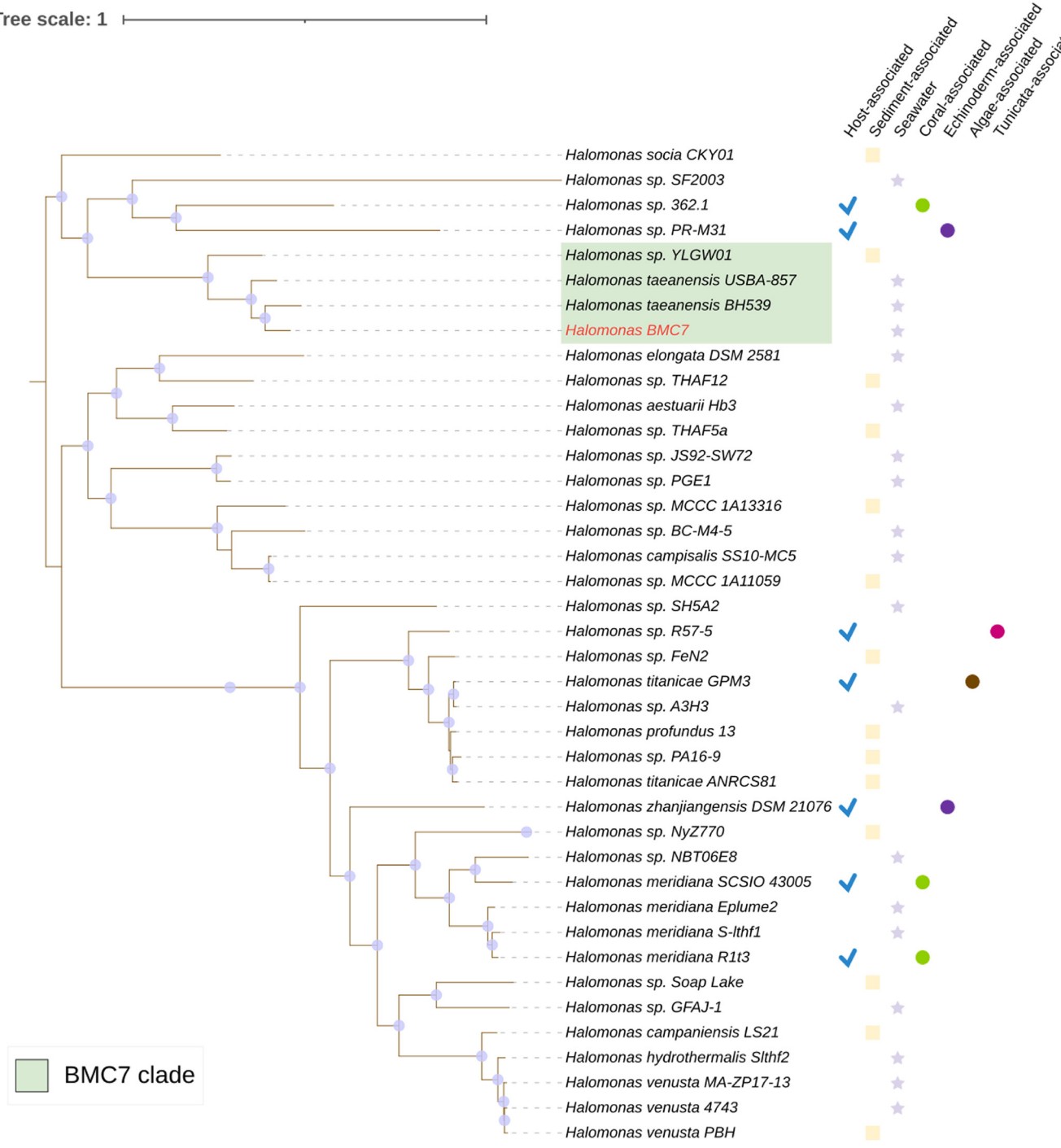

**FIG 2** Phylogenomic inference of publicly available genomes from 39 *Halomonas* strains and the genomes of BMC7 strain (in red), totaling 40 genomes. The green-shaded area represents the strains that form a monophyletic clade with BMC7. The tree was assembled from the comparison of 1,000 proteins through the codon tree method of the PATRIC platform that selects global protein families (PGFams) as homology groups and analyzes aligned proteins and DNA encoding single-copy genes using the RAxML program. The best protein model found by RAxML to build this tree was LG. Purple dots on the branch length correspond to the bootstrap value of 100. The icons on the right side of the strains represent their isolation sources.

BMC strains 1 to 5, which formed a BMC clade (Fig. 3). A total of 1,076 unique BMC genes were identified in this pangenome (i.e., genes that were present only in the genome of one or more *P. shioyasakiensis* BMCs 1 to 5, but not in the genomes of the other 17 *Pseudoalteromonas* strains). Of these, the functions of only approximately one-third (208 genes) are known, while two-thirds (868 genes) were classified as hypothetical proteins. Among the 208 genes with known functions, 8 were shared by all *P. shioyasakiensis* BMC

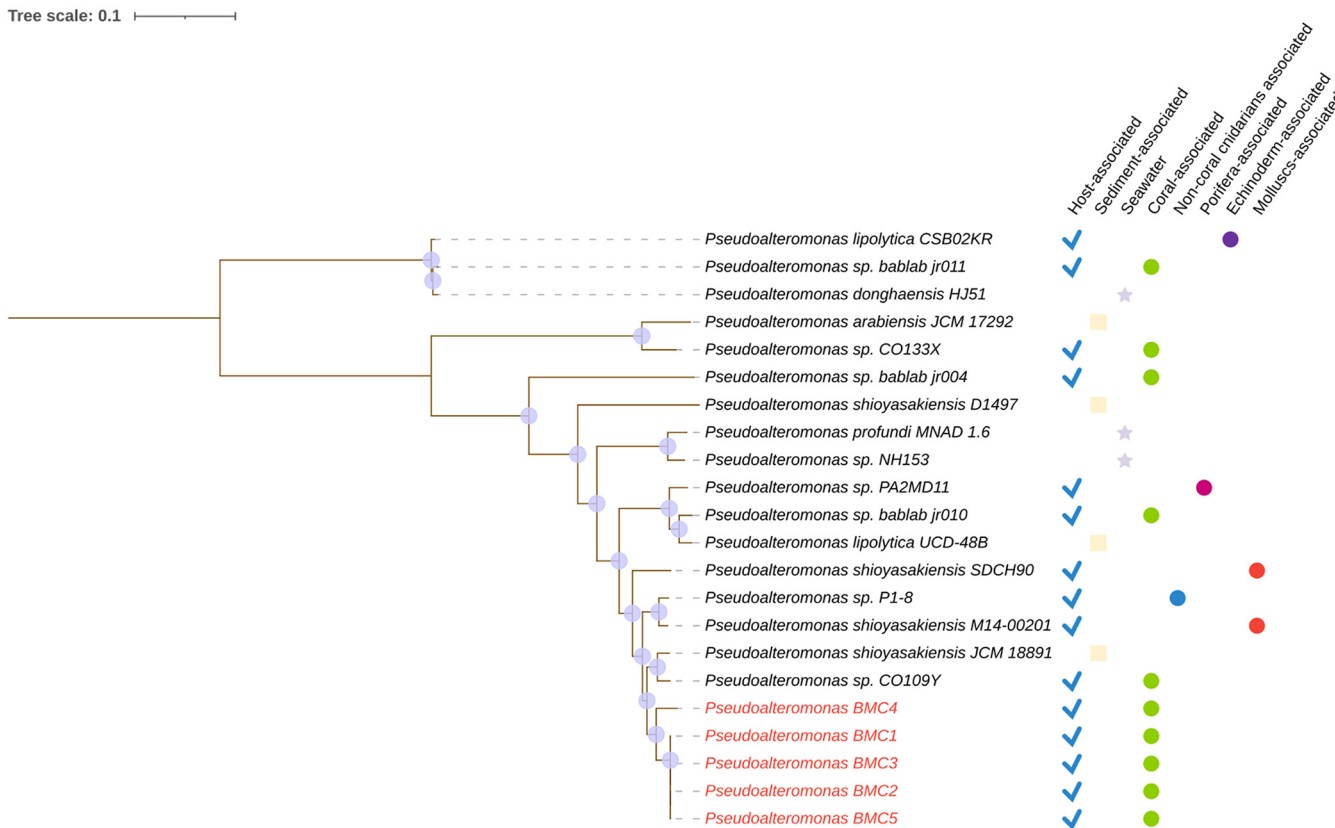

**FIG 3** Phylogenomic inference of publicly available genomes from 17 *Pseudoalteromonas* strains and the genomes of BMC strains 1 to 5 (in red), totaling 22 genomes. The tree was assembled from the comparison of 1,000 proteins through the codon tree method of the PATRIC platform that selects global protein families (PGFams) as homology groups and analyzes aligned proteins and DNA encoding single-copy genes using the RAxML program. The best protein model found by RAxML to build this tree was Jones-Taylor-Thornton - Direct Computation with Mutabilities (JTTDCMut). Purple dots on the branch length correspond to the bootstrap value of 100. The icons on the right side of the strains represent their isolation sources.

strains, 49 were present in four BMC strains, 7 genes were found only in three BMC strains, 1 gene was detected only in two BMC strains, and 143 genes were exclusive to BMC 4 (Fig. 4). Of the 208 unique BMC genes with known functions, 2 encoded previously proposed beneficial traits for corals (Table S6), both related to iron bioavailability.

A total of 382 singleton genes were identified for *Cobetia* sp. BMC 6, compared to the other 10 *Cobetia* genomes used (Fig. 5). Of these, 162 genes were known, while 220 were classified as hypothetical proteins. Six of the known genes encoded previously defined beneficial traits for corals (Table S6) related to sulfur cycling, iron bioavailability, antibiotic production, vitamin B synthesis, nitrogen cycling, and oxidative stress protection.

The phylogenomic assessment indicated that *Halomonas* sp. BMC 7 formed a clade with the other three *Halomonas* sp. strains used for the pangenome analysis (Fig. 2). A total of 1,447 genes were identified as singletons, which were detected only in the genome of *Halomonas* sp. BMC 7 (Fig. 6). Among the 1,447 genes, 737 genes encoded hypothetical proteins, while 710 genes have a known function. Among these 710 genes with known functions, 18 genes encode previously proposed beneficial traits for corals (Table S6) related to siderophores production, vitamin B complex synthesis, sulfur cycling, response to oxidative stress, nitric oxide detoxification, and nitrogen cycling, among others. Fig. 7 summarizes the beneficial features of BMC singleton genes for coral and algal endosymbionts.

## DISCUSSION

***Cobetia* sp. BMC 6 and *Halomonas* sp. BMC 7 are new species candidates.** ANI and DDH values, together with our phylogenomic inference, confirmed that BMC strains 1 to 5 belong to the same species, namely, *P. shioyasakiensis*, which was initially

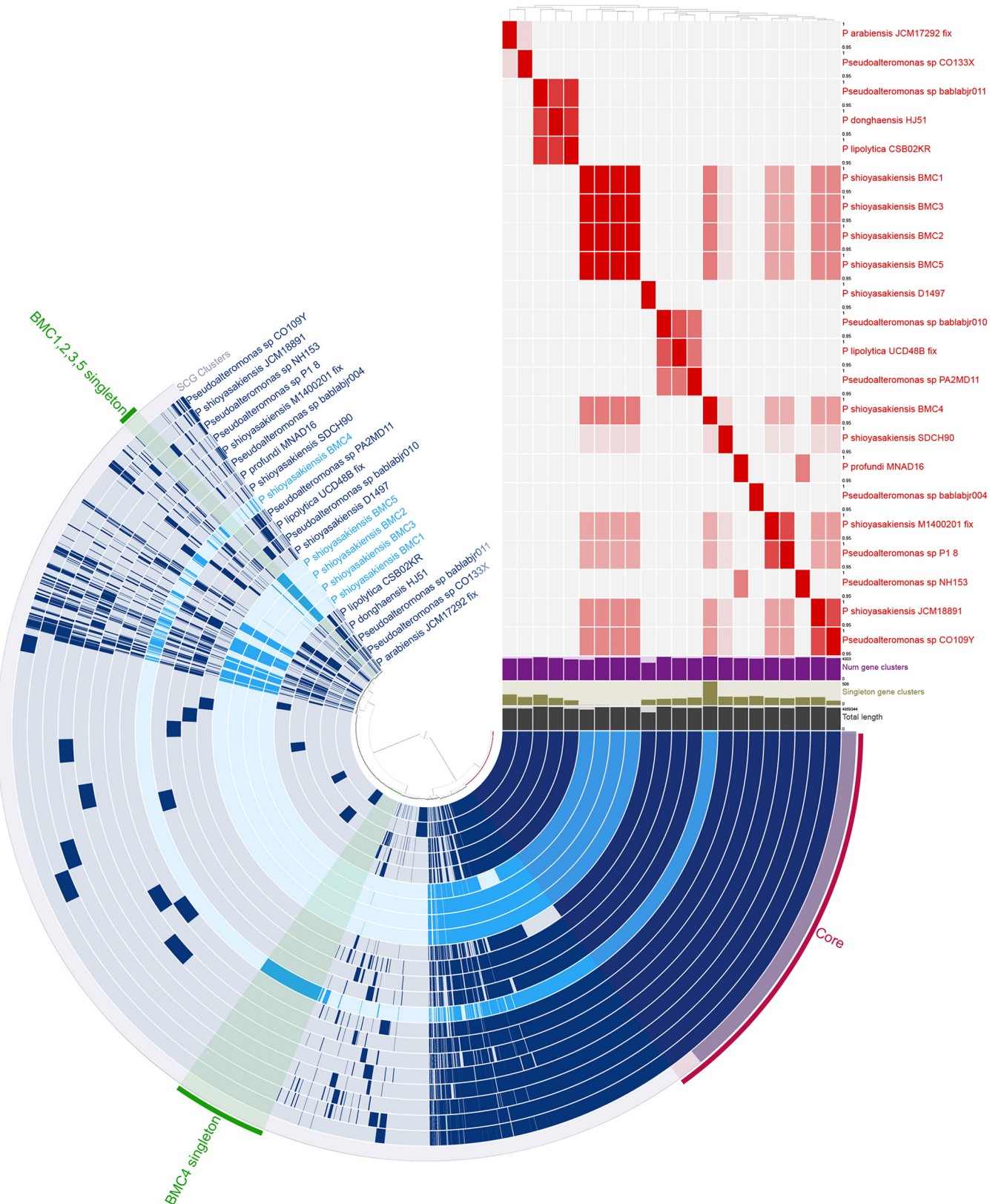

**FIG 4** Pangenomic analysis of *Pseudoalteromonas* strains. The outermost circle displays the single-copy genes (SCG cluster) that were shared by all the genomes. The presence and absence of coding sequences (CDS) in the genome are indicated in dark blue and light blue, respectively. The average nucleotide identity (ANI) (95% to 100%) comparison among all 22 genomes is displayed with the red and white heat map. The dendrogram at the top of the heat map represents the hierarchical clustering of genomes based on the occurrence of gene clusters. The layers underneath the %ANI heat map, from top to bottom, indicate number of gene clusters (purple), number of singleton gene clusters (dark green), and total length (gray) of each genome.

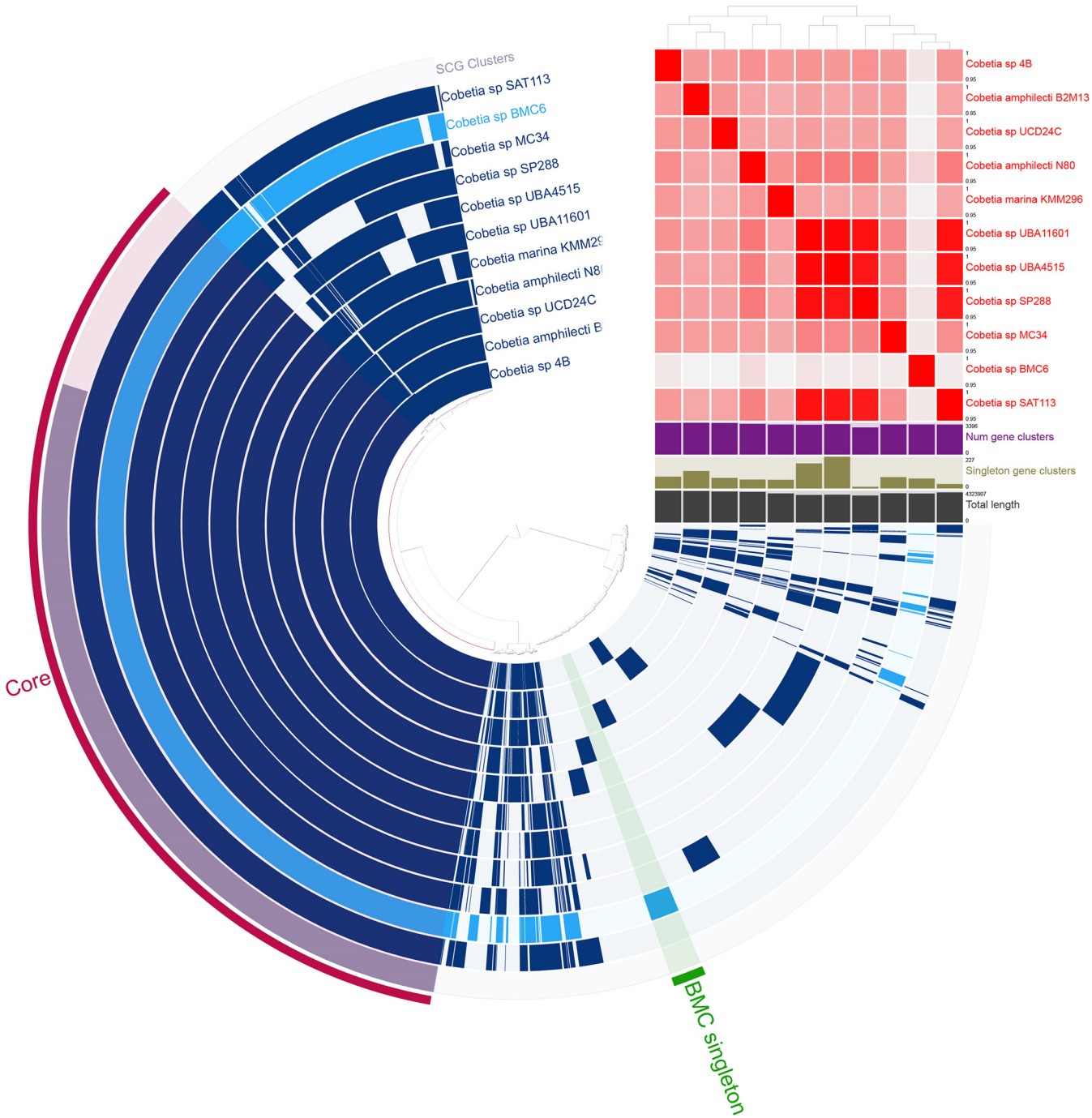

**FIG 5** Pangenomic analysis of *Cobetia* strains. The outermost circle displays the single-copy genes (SCG cluster) that were shared by all the genomes. The presence and absence of coding sequences (CDS) in the genome are indicated in dark blue and light blue, respectively. The ANI (95% to 100%) comparison among all 11 genomes is displayed with the red and white heat map. The dendrogram at the top of the heat map represents the hierarchical clustering of genomes based on the occurrence of gene clusters. The layers underneath the %ANI heat map, from top to bottom, indicate number of gene clusters (purple), number of singleton gene clusters (dark green), and total length (gray) of each genome.

described and isolated from marine sediment (43). The difference in genome size and ANI of BMC 4 compared to BMCs 1, 2, 3, and 5 indicates that BMC 4 may be a more divergent strain or even a subspecies of *P. shioyasakiensis*.

The higher similarity between the genomes of BMC strains 6 and 7 compared to BMCs 1 to 5 was expected as the two genera, *Cobetia* and *Halomonas*, respectively, belong to the *Halomonadaceae* family and have only recently been separated into two

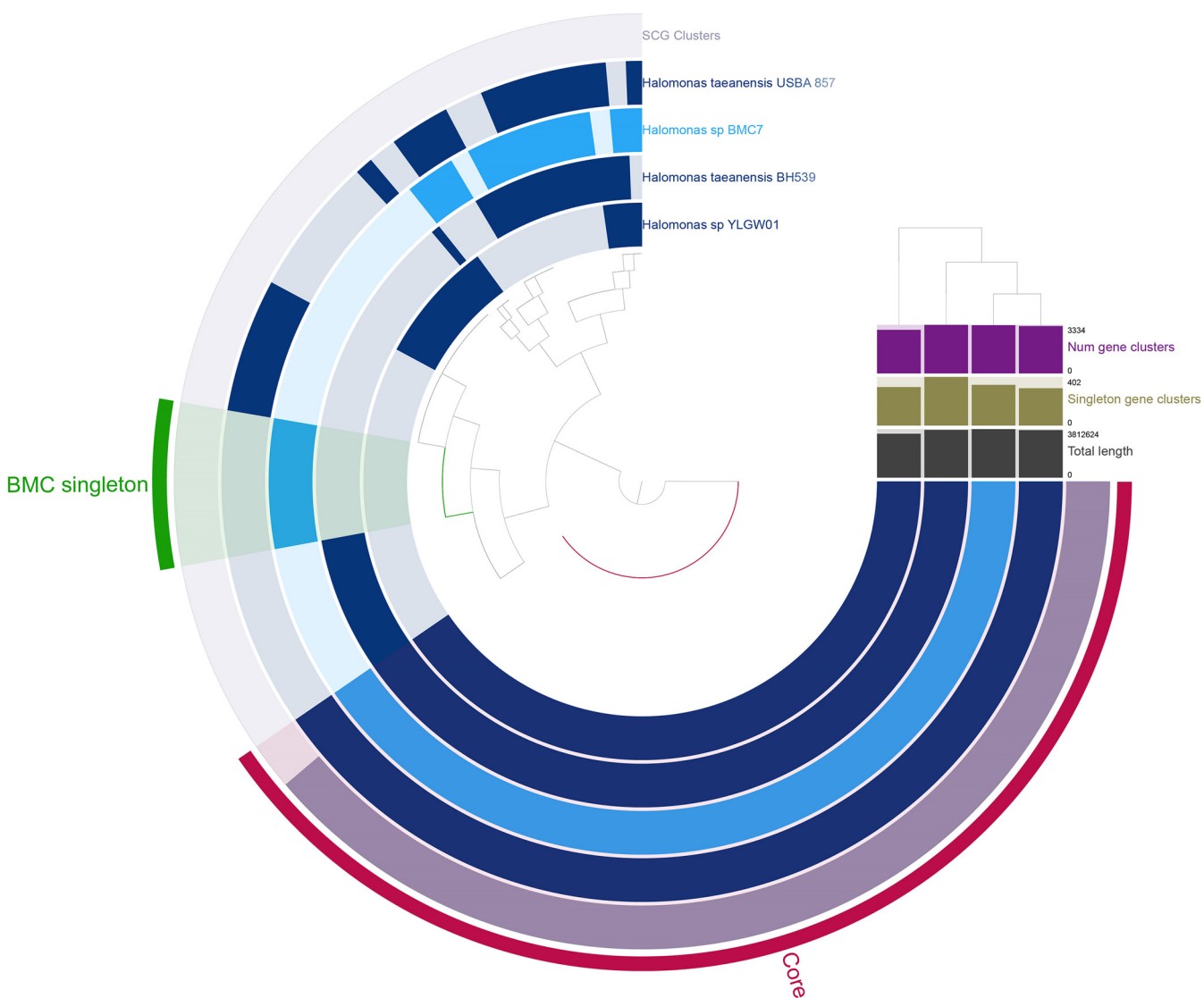

**FIG 6** Pangenomic analysis of *Halomonas* strains. The outermost circle displays the single-copy genes (SCG cluster) that were shared by all the genomes. The presence and absence of coding sequences (CDS) in the genome are indicated in dark blue and light blue, respectively. The dendrogram at the top of the number of gene clusters represents the hierarchical clustering of genomes based on the occurrence of gene clusters. The layers underneath the dendrogram, from top to bottom, indicate number of gene clusters (purple), number of singleton gene clusters (dark green), and total length (gray) of each genome.

distinct genera (44). Unlike the taxonomic and phylogenomic results obtained for the *Pseudoalteromonas* BMCs 1 to 5, the results of *Cobetia* sp. BMC 6 and *Halomonas* sp. BMC 7 did not allow the classification of these strains at the species level. With the ANI taxonomic attribution method, *Cobetia* sp. BMC 6 showed a similarity higher than 95% with the closest 10 strains (Table S2) that form a clade in the *Cobetia* sp. phylogenomic tree (Fig. 1). However, the GGDC index indicates the highest similarity value of 64.10% with *Cobetia amphilecti* N80, which is below the cutoff value of 70% defined for the classification of the same species. Considering these results together, the species definition for this BMC is inconclusive, and therefore, we suggest the use of *Cobetia* sp. when referring to this BMC strain. *Cobetia* sp. BMC 6 may represent a new species within the genus *Cobetia*, but further analyses, such as phenotypic and biochemical tests related to the genus and API tests, are necessary to determine this (45).

Although a phylogenomic clade was formed by *Halomonas* sp. BMC 7 strain with *Halomonas* sp. YLGW01 and *H. taeanensis* strains, both ANI and DDH values were below the defined species-level thresholds, and we also suggest the use of *Halomonas*

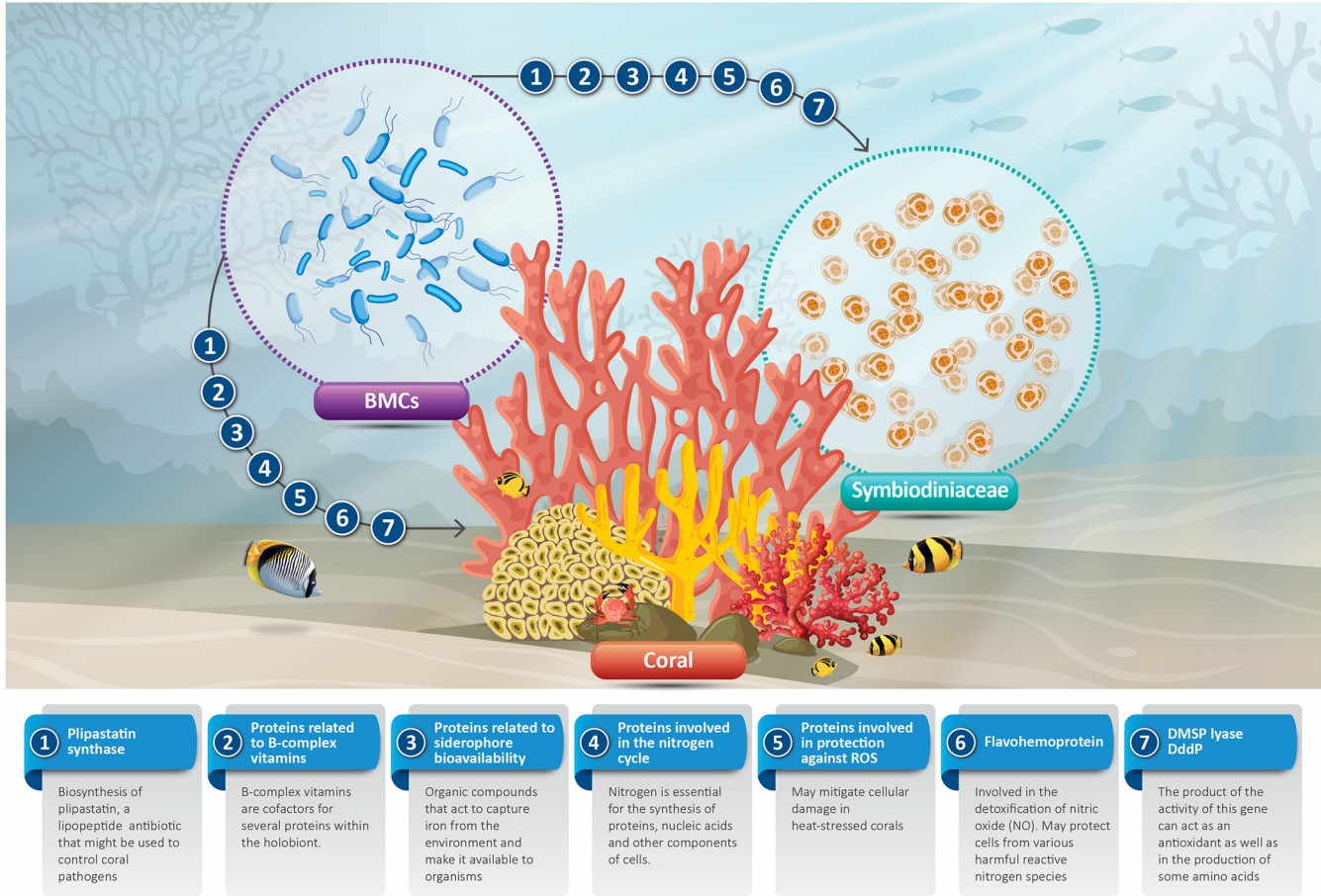

**FIG 7** Theoretical illustration of the benefits that singleton genes of BMCs might bring to the coral host and its endosymbiotic algae.

sp. when referring to BMC 7. This BMC is possibly a new species of *Halomonas*, but further analyses to characterize the species are needed, as described for *Cobetia* sp. BMC 6. Based on this evidence, the lack of comparative genomes likely contributed to the low taxonomic resolution of both the *Cobetia* and the *Halomonas* BMCs.

**Genome analyses confirm the presence of previously suggested BMC mechanisms.** Following the phylogenomic and taxonomic inference of the BMCs, we searched their genomes for genes encoding proteins that could be potentially beneficial for corals. These genes are not exclusive to BMCs and can also be found in any non-BMC strains. In addition to genes encoding catalase and nitrite reductase enzymes identified via PCR and biochemical tests, which were used for their selection as potential BMCs (7), genes related to other putative beneficial traits were also detected. These included several genes encoding proteins with antioxidant roles, such as superoxide dismutase (present in all BMCs). Superoxide is a product of oxygen metabolism and can cause various types of cell damage (46). Another important gene associated with the antioxidant role found in all BMCs is the gene named *GSS*, which encodes the protein glutathione synthetase and is related to the glutathione biosynthesis pathway (47); glutathione can be used by the protein glutathione peroxidase, also present in all BMCs, to neutralize reactive oxygen species (ROS), such as $H_2O_2$ (48). $H_2O_2$, among other ROS, are produced in coral holobionts during periods of high temperatures and irradiance, causing damage to host and symbiont cells (49–51). The production of ROS is directly linked to coral bleaching (52, 53). ROS-scavenging BMC strains have been hypothesized to alleviate bleaching signs in stressed coral holobionts (26) and should therefore be targeted for BMC selection.

Genes associated with the synthesis of cobalamin, also known as vitamin $B_{12}$, a cofactor involved in the production of the amino acid methionine (which is necessary to synthesize all proteins), were observed in all BMC genomes, except in *Cobetia* sp. BMC 6. Vitamin $B_{12}$ biosynthesis is associated with several metabolic pathways, including the generation of glutathione and DMSP antioxidants (54), which are mechanisms that can directly help corals during heat stress by mitigating increased concentrations of ROS (26, 41), and has therefore been recently suggested as a BMC trait (41). In addition, Symbiodiniaceae is not capable of synthesizing cobalamin, so adding BMCs that may play this role would help to meet the needs of endosymbiotic algae. Again, this is theory that is described in the literature (e.g., Peixoto et al. [26, 41], Matthews et al. [42], and Santoro et al. [10]); several more specific studies are needed to substantiate this theory. Some such studies are already being performed by our laboratory, as well as in other laboratories. We believe the caveats discussed above will be important to highlight that this is a roadmap for the screening of BMCs that will be then validated using manipulative studies followed by multiomics and physiological analyses, as for example in the work of Santoro et al. (10). Algae belonging to the *Symbiodiniaceae* family require exogenous cobalamin for growth, because they do not have the genetic machinery to generate the active form of this vitamin (42), implying that *Symbiodiniaceae* rely on bacterial symbionts to access this important cofactor.

Both *Cobetia* sp. BMC 6 and *Halomonas* sp. BMC 7 also carry different genes associated with the synthesis of siderophores and ectoines (55). Siderophores are organic compounds that act to capture iron from the environment, and their production has also been suggested as a BMC trait (41). Iron is essential for several physiological processes in corals and microalgae, including photosynthesis, respiration, and nitrogen fixation (56), yet bioavailable iron concentrations in most of the global oceans are too low to support growth of microalgae (57). Several marine bacteria produce siderophores, which bind and concentrate iron in bioavailable forms, allowing the absorption of this limiting micronutrient by phytoplankton. For instance, the production of siderophores by the gammaproteobacterium *Marinobacter* promotes the growth of its dinoflagellate partner *Scrippsiella trochoidea* (58). *Marinobacter* strains are also a component of the "core microbiome" shared between different cultures of *Symbiodiniaceae* (59), which likely rely on bacterial partners to meet their bioavailable iron needs.

Ectoines, which include the compounds ectoine and hydroxyectoine, have several roles in the physiology of prokaryotic and eukaryotic organisms, including protection against radiation and oxidative stress, enzyme stabilization, and cryoprotection during freeze-thaw processes. Bownik et al. (60) tested the potential role of ectoines as thermal protectors in *Daphnia magna*, a small planktonic crustacean. Their results showed protective effects of ectoine in *D. magna* subjected to heat stress, which was the first indication of the beneficial effects that ectoines can generate in aquatic animals undergoing this type of stress. Therefore, we hypothesize that this type of protection can also occur in corals undergoing thermal stress, and bacteria that synthesize ectoines are potential targets for selection as BMCs. We also suggest that ectoine production may have been one of the protective mechanisms promoted by BMCs 6 and 7 in the work of Rosado et al. (7).

Furthermore, we detected genes whose functions are involved in the synthesis of glycine betaine (GB) as a common gene across our BMC strains. GB acts as a compatible solute (osmolyte) to neutralize the effects of high salinity and contributes to the protection of membranes and proteins against abiotic stressors (61, 62). Ngugi et al. (63) suggested that GB metabolism acts on coral physiology and symbiotic interactions. The authors highlighted the fact that GB comprises about 16% of the total nitrogen biomass of corals, being also a potential nitrogen source for the holobiont. In addition to its properties as an osmolyte, GB may also have additional properties that replace other osmolytes, such as DMSP. In plants, for example, GB inhibits heat and irradiance stress by stabilizing vital extrinsic proteins of the photosystem II complex (61). A similar role is hypothesized in corals and their endosymbiotic algae (64), as shallow water corals and those exposed

to high irradiance have higher concentrations of GB compared to colonies that live in deeper water and shaded areas (64). Finally, GB metabolism may be a key determinant of carbon metabolism that supports methyl-dependent processes (such as DNA methylation and protein synthesis). As carbon and nitrogen metabolism are often closely linked, it is hypothesized that the symbiosis between corals and their carbohydrate-producing algal symbionts may have selectively enhanced GB sequestration and storage to support metabolic activity, maintaining a proper osmotic environment while serving as a nitrogen reservoir to accommodate the growing needs of endosymbionts (63). It is important to note that these studies did not directly test the role of GB in heat-stressed corals; thus, future research addressing this topic is required. For these reasons, in addition to the high abundance of GB-producing strains in our BMC consortium, we also suggest GB production as a BMC mechanism that was likely associated with the successful application of the consortium by Rosado et al. (7).

The BMC traits mentioned above bring direct benefits to corals and/or their endosymbiotic algae and indicate potential mechanisms that may have contributed to the bleaching mitigation observed (7), in addition to the characteristics previously used to select the BMC consortium. In some cases, they also represent new BMC targets that should be investigated in future research efforts (such as the production of different ROS scavengers, GB, and ectoines). Furthermore, transcriptomic and/or metabolomic surveys will always be required in thermal stress experiments to confirm whether these BMC mechanisms are being expressed and correlated with thermal protection (10).

**Pan-(BMC)genomes suggest additional, novel BMC mechanisms.** We also generated pangenomes for the BMCs and other non-BMC strains to identify genes that are unique to the selected BMC strains. The *Pseudoalteromonas* pangenome identified 208 genes with known functions that have been identified only in the BMC genomes (i.e., in one or more strains of the *P. shioyasakiensis* BMCs 1, 2, 3, 4, and 5) (Fig. 4). Among those genes, we identified two encoded proteins that may have a putative beneficial role, related to iron bioavailability, for corals and *Symbiodiniaceae*: ferri-bacillibactin esterase BesA (also found in *Cobetia* sp. BMC 6) and luminescence regulatory protein LuxO. The ferri-bacillibactin esterase hydrolyzes the ferri-bacilibactin (ferri-BB) complex during the trilactone cycle, leading to the release of cytosolic iron, making iron available for metabolic use (65). LuxO plays a role in the production of siderophores (66). As mentioned previously, iron is essential for a number of physiological processes in corals and microalgae, including photosynthesis, respiration, and nitrogen fixation (56). Notably, more than two-thirds of the genes found exclusively in *P. shioyasakiensis* BMCs are classified as hypothetical proteins of unknown function, and several of these genes may or may not play a vital role in the holobiont health.

The *Cobetia* pangenome identified 162 genes with known functions that were identified in the *Cobetia* sp. BMC 6 genome only (Fig. 5). Among these genes, six encode proteins that may play a putative beneficial role for corals and *Symbiodiniaceae*. These proteins include plipastatin synthase, an enzyme involved in the biosynthesis of plipastatin, an antimicrobial lipopeptide that has been reported to have a broad antagonistic effect on various soil and postharvest fungal phytopathogens, specifically on filamentous fungi (67), in addition to possessing antibacterial and antiviral activity (68–70). These genes represent specific BMC mechanisms involved in the biological control of pathogens in coral (26). The protein cyclic pyranopterin monophosphate synthase was also exclusive to the *Cobetia* sp. BMC 6 genome. This protein is involved in the biosynthesis of molybdopterin, an important cofactor of some enzymes that represent putative BMC mechanisms (i.e., dimethyl sulfoxide [DMSO] reductase and nitrate reductase) (71). Another protein exclusive of the *Cobetia* sp. BMC 6 genome is heme A synthase, a component of many biologically important hemoproteins, which include proteins related to protection against oxidative stress, such as catalases and heme peroxidase (72).

The protein siroheme synthase is encoded by the *cysG* gene and was present only in the genomes of *Cobetia* sp. BMC 6 and *Halomonas* sp. BMC 7, being absent in the other genomes used in the pangenome analyses of both genera. This may have occurred

because there were limited data for these pangenome analyses, particularly for the *Halomonas* genus, for which there were only three representative genomes. This protein is involved in sulfur and nitrogen metabolism (73, 74), both of which are important for the coral holobiont and were proposed as BMC traits (41). In addition, three genes encoding proteins related to the nitrogen cycle (nitrite reductase, nitrate reductase, and urease protein) were found only in the genomes of *Cobetia* sp. BMC 6 (nitrite reductase) and *Halomonas* sp. BMC 7 (all three genes). BMCs associated with the biogeochemical cycling of nitrogen are potentially key members of tropical coral microbiomes, as coral reef systems are characterized by high primary productivity and low nutrient availability (75, 76). To thrive in oligotrophic environments, corals rely on the rapid assimilation and retention of nitrogen, which is the major limiting element of primary production in the ocean (77), whereas excessive amounts of nitrogen can also represent a threat to corals (78). Therefore, BMCs involved in nitrogen cycling may play an important role in alleviating nutrient limitation while also cycling excessive amounts of nitrogen compounds, helping to maintain host tolerance to nutrient-dense conditions such as those experienced during seasonal flood events (79) or helping to maintain a favorable nitrogen-phosphorus ratio (80).

A total of 710 genes with known functions were identified in the *Halomonas* sp. BMC 7 genome only (Fig. 6). Among those genes, 18 encode proteins were identified that may play a beneficial role for corals and *Symbiodiniaceae*, such as those related to nitrogen cycling and the siroheme synthetase protein mentioned above. *Halomonas* sp. BMC 7 harbors a gene encoding the protein flavohemoglobin, an oxidoreductase enzyme involved in the detoxification of nitric oxide (NO) in an aerobic process called the nitric oxide dioxide (NOD) reaction. This reaction uses $O_2$ and NADPH to convert NO to nitrate and in this way protects cells from various harmful nitrogen compounds (81). It is worth highlighting that NO demonstrates a wide range of cellular toxicities that are known as "nitrosative stress," an analogous term to oxidative stress. Although NO drives some of these toxicities directly, it is also increasingly clear that NO serves as a precursor molecule for a variety of reactive nitrogen species (RNS), such as dinitrogen trioxide ($N_2O_3$), nitrogen dioxide ($NO_{2\bullet}$), and peroxynitrite ($ONOO_-$) (82, 83). These highly reactive RNS drive many biochemical reactions involving cellular proteins (e.g., nitration, nitrosylation, oxidation), DNA (e.g., deamination), and lipids (e.g., nitration and oxidation). The final product is usually loss of function for this affected biomolecule (84, 85). During heat stress, the amount of NO inside coral cells and *Symbiodiniaceae* increases, while excess NO reacts with $O_2$ to form peroxynitrite, which in turn inhibits the transport of electrons in the mitochondrial electron transport chain (49, 86) and interrupts the host's cellular respiration. Increased nitric oxide reductase and arginase-related genes (both involved in supporting cells in detoxifying nitric oxide from the host) have been previously correlated with diseased coral tissue (87). Therefore, any mechanism that reduces the excess of intracellular NO in the holobiont seems to be important for the health of corals.

Six genes encoding proteins related to the synthesis of B vitamins were found in *Halomonas* sp. BMC 7 only; these were *cobP*, *cobD*, and *cobQ* (related to cobalamin [$B_{12}$] synthesis); the *thiL* gene related to thiamin ($B_1$) synthesis; the *panC* gene related to pantothenate ($B_5$) synthesis; and the *folC* gene related to folate ($B_9$) synthesis. The production of B vitamins has been observed in other relationships between corals and bacteria and may be an important process for healthy coral functioning (88). The micronutrient cobalamin (vitamin $B_{12}$) is involved in diverse metabolic pathways, including the generation of the antioxidants glutathione and DMSP (54). Thiamin ($B_1$) plays a key role in intermediate carbon metabolism in algae and is a cofactor for several enzymes involved in the metabolism of primary carbohydrates and branched-chain amino acids (89). Folate ($B_9$) is involved in the synthesis of *S*-adenosylmethionine, a methyl group donor in DNA methylation (90). Downregulation of genes involved in the pantothenate ($B_5$) metabolic process has been observed in corals 10 h after heat stress, and this has been hypothesized to increase host susceptibility to pathogens due to downregulation of innate immune response (91). Together, these data reinforce that the production of B-complex

vitamins is an important BMC characteristic. It is also possible that *Symbiodiniaceae* obtain B vitamins from the bacterial symbionts, as many algae do not have the capacity to produce these vitamins (89). Indeed, a recent study showed that the *Gammaproteobacteria* symbionts of healthy octocoral hold the genetic blueprint for the biosynthesis of all eight B vitamins (92).

Three genes encoding proteins related to siderophore biosynthesis were identified in *Halomonas* sp. BMC 7 only (*iucC*, *iucD*, and *iucA*). As previously mentioned, siderophores are organic compounds that capture iron from the environment; they bind and concentrate iron in bioavailable forms, allowing the absorption of this limiting micronutrient by many organisms. Another three genes related to protection against oxidative stress were also identified in the *Halomonas* sp. BMC 7 genome (*katG*, *oxyR*, and *ubiG*). An increase in temperature and irradiance results in increased production of ROS in corals and their endosymbiont photosynthetic algae symbionts, as discussed earlier, which overloads the defense mechanisms of both players and ultimately damages their cells so that it is one of the main drivers of coral bleaching (49–51).

A gene related to the DMSP degradation pathway was also identified in the *Halomonas* sp. BMC 7 genome only. The *dddP* gene encodes the protein dimethlysulfoniopropionate lyase, the enzyme that breaks down DMSP. The resulting decomposition products can act as antioxidants and protect algae from oxidative stress derived from photosynthesis (93, 94). Furthermore, Garren et al. (95) showed that the pathogen *V. coralliilyticus* displays DMSP chemotaxis to locate corals stressed by increasing temperature, wherefore the DMSP catabolism could be a mechanism that makes it difficult for the pathogen to localize the coral. In addition, a probiotic BMC consortium using a DMSP degrading strain has significantly reduced DMSP concentrations within the holobiont, which was also correlated with significant improvements on coral health and survivorship (10). Therefore, genes related to DMSP metabolism would fit in as an excellent target to search for BMC strains.

Many studies have highlighted the vital role of the microbiome and its metabolites in maintaining coral health and regulating ecosystem resilience (10, 28, 30). Microorganisms can mitigate anthropogenic impacts through their role in holobiont regulation (7, 20), as well as disrupting nutrient and energy flows in coral reefs (26, 96). It is therefore of great importance to understand the specific mechanisms of interactions between corals and their associated microbes. Although this study did not aim to show that BMCs have certain characteristics that that are not present in other members of the coral microbiome, our approach may facilitate the detection of microorganisms harboring these theoretically vital roles to the host health, which could therefore be tested as BMCs. The best way to validate the use of a BMC consortium as a means to mitigate damage caused by different stressors is to compare any health improvement with the use of a placebo treatment. Alternative microbial therapies, such as the use of heat-killed bacteria (also called postbiotics) can also help to understand the protective mechanisms provided by microbes (dead or alive). The use of microbes that were not selected by current screening methods can also generate data on unknown beneficial mechanisms.

In summary, our results indicate several possible theoretical ways in which BMCs could have acted to help corals during periods of stress (7). Although, as highlighted above, most of these characteristics are still theoretical, some of the putative beneficial genes have already been validated or shown to have differential expression during heat stress events in corals. For instance, metatranscriptome analyses have shown the differential expression of several genes in response to thermal stress in scleractinian corals, such as *Mussismilia hispida*, *Siderastrea radians*, *Orbicella faveolata*, and *Pseudodiploria clivosa* (10, 97). The results reveal that several genes previously suggested to be potentially beneficial to corals (and also detected in our genomes) were upregulated during stress, such as peroxidase genes related to protection against ROS (e.g., *S. radians* and *O. faveolata*), vitamin B biosynthesis (e.g., *P. clivosa*) (97), and host immune response elicited by shifts in the microbiome (10). Genes related to sulfur and nitrogen metabolism were also upregulated, with a higher expression in *S. radians*. This differential expression was mainly due to shifts

in the microbiome, suggesting that the microbiome is primarily responsible for the upregulation of these genes in the studied corals (10, 97).

Considering the limitations and necessary improvements for the cultivation of coral-associated microbes (98), the use of *in silico* screenings can simultaneously accelerate the selection of cultured BMCs (since through genome analysis it is possible to quickly identify a greater variety of genes compared to PCR and biochemical/physiological assays) and improve our ability to culture alternative BMCs. This work provides an array of new gene targets to be incorporated into the BMC screening framework, while it also suggests potential alternative beneficial mechanisms involved in the microbial mitigation of coral bleaching.

## MATERIALS AND METHODS

**Genome sequencing of the BMC consortium.** Seven bacterial strains isolated from the coral *P. damicornis* by Rosado et al. (7) were originally selected, based on their putative beneficial traits (26), for a mesocosm experiment that demonstrated their beneficial effects on corals (7). These strains were previously identified by 16S rRNA gene sequencing as *Pseudoalteromonas* sp. (*n* = 5), *Cobetia* sp. (*n* = 1), and *Halomonas* sp. (*n* = 1).

For genome sequencing, the seven BMC strains were cultivated in Difco marine broth 2216 culture medium (Becton, Dickinson & Co. Sparks, USA) from their respective stocks kept in glycerol at −80°C. After 28 h under constant agitation at 26°C to 28°C, 1 mL of each strain was centrifuged at 13,000 × g for 2 min in 1.5-mL tubes; the supernatant was discarded, and the cell pellet was used for DNA extraction. For this purpose, a Wizard genomic DNA purification kit (Promega, Madison, WI, USA) was used following the manufacturer's protocol. The genomic DNA was quantified using a Qubit 2.0 fluorometer double-stranded DNA (dsDNA) kit (Invitrogen, Carlsbad, CA, USA) and subjected to agarose gel electrophoresis (1%) to observe DNA quality. Genome sequencing of the seven strains was performed on an Illumina MiSeq platform with the NEBNext Ultra II FS DNA kit for library assembly and a MiSeq reagent kit v3 for flow cell, which generated between 1,405,296 and 2,187,986 paired end reads of 301 bp/genome for a total of 600 amplification cycles.

**Processing and assembly of the BMC genomes.** Raw sequence reads were quality-filtered using the Trimmomatic version 036 program (99) with a Phred score parameter value of 33 and the following arguments: sliding window size: 4; sliding window minimum quality: 20; head crop length: 5; leading minimum quality: 10; trailing minimum quality: 10; and minimum read length: 36. Quality of the filtered sequences was evaluated by the FastQC v. 0.11.5 program (http://www.bioinformatics.babraham.ac.uk/projects/fastqc/), followed by assembly of each genome using SPAdes v. 3.13.0 (100) with default parameters. Genome integrity and contamination values of the assembled genomes were calculated with the CheckM software version 1.0.18 (101), and sequences with a high contamination index were refined using the RefineM program version 0.0.23 (101, 102). Stretches with discrepant taxonomy sequences were removed using RefineM's default settings, which allowed for coverage and contamination indices to be recalculated.

**Taxonomic attribution analysis.** Species-level classification of the seven genomes was performed using the FastANI tool (103) and the Genome-to-Genome Distance Calculator (GGDC). FastANI makes a pairwise comparison of complete genomes, but instead of aligning the entire sequence of the two genomes, it uses the Mashmap program (104). ANI values of >95% are widely used as a cutoff to determine the same-species status of genome sequences (103, 105, 106). The GGDC is a state-of-the-art *in silico* method for genome-to-genome comparison, reliably mimicking conventional DNA-DNA hybridization (DDH). A GGDC index with more than 70% similarity between two genome sequences indicates that both sequences belong to the same species. A DDH above 79% suggests that a pair of sequences belongs to the same subspecies (105, 107, 108). For the identification of BMCs 1, 2, 3, 4, and 5, previously identified by 16s rRNA gene analysis as *Pseudoalteromonas* sp. (7), the 17 genomes that formed a cluster with the BMCs (as assessed in a preliminary analysis in Fig. S1) were used as illustrated in the *Pseudoalteromonas* phylogenomic trees (Fig. 3; Fig. S1). For the identification of BMC 6, 10 genomes that formed a cluster with the BMC were used as illustrated in the *Cobetia* phylogenomic tree (Fig. 1). Finally, to identify BMC 7, three genomes that formed a cluster with the BMC were used as illustrated in the *Halomonas* phylogenomic tree (Fig. 2).

**Structural and functional annotation of genomes.** The seven genomes were annotated using Prokka (109), as well as the PATRIC Bacterial Bioinformatics Resource Center (110), which uses the RASTtk online server (111) with default parameters. The output files were visualized on PATRIC's graphical user interface to build genomic maps of the studied bacteria and search for protein-coding genes indicating potentially beneficial characteristics for corals, such as genes related to nutrition and growth, mitigation of toxic compounds or stress, early coral development, and pathogen control (26, 41). The genes of interest were found by searching the subsystems generated by the RASTtk platform in PATRIC from the databases used in their default configurations.

**Assembly of phylogenomic trees.** To build phylogenomic trees, PATRIC's codon tree method was used, which randomly selects up to 10 proteins to represent each genus-level family and combines them to form a single set of representatives in order to prevent cluster formation that is based upon the genus rather than protein similarity (112). Both protein (amino acid) and gene (nucleotide) sequences were used for each of the selected genes from the PATRIC global protein families (PGFams). Protein sequences were

aligned using MUSCLE (113), and nucleotide-encoding gene sequences were aligned using the Codon_align function of BioPython (114). A file in phylip format was generated with the concatenated alignment of all proteins and nucleotides, and then a partitioned file for RAxML (115), describing the alignment in terms of proteins, was generated. Support values were generated using 100 rounds of the "Rapid" bootstrapping option (116) in RAxML. The phylogenomic tree was formed by concatenated open reading frames (ORFs) from 1,000 protein-coding genes (Fig. 1 to 3) and 100 protein-coding genes (Fig. S1) shared among all genomes present using the best protein model method found by RAxML (117). The result was generated in Newick format and visualized in iTOL version 6.5.3 (118), where the tree was assembled.

Three different trees were generated, one for each genus of BMC (*Pseudoalteromonas*, *Cobetia*, and *Halomonas*). The genomes of each genus that made up the trees were selected from the PATRIC database using different selection criteria, such as being reference genomes and/or complete genomes assembled in one contig of bacteria isolated from corals and/or aquatic environments. The host from which each strain was isolated was also added to the phylogenomic tree.

**Pangenomic analysis and gene categorization of BMC genomes.** The pangenome was generated separately for each genus of BMC (*Pseudoalteromonas*, *Cobetia*, and *Halomonas*) through the Roary program (119), using the default settings based on annotations made with Prokka. To generate each pangenome, sequences from the closest genomes of all BMCs based on taxonomic attribution results and phylogenetic trees were added to the analysis. More specifically, in the pangenome of the *Pseudoalteromonas* genus, 17 genomes of *Pseudoalteromonas* sp., which formed a clade with the BMC (Fig. S1), were added in addition to the 5 *Pseudoalteromonas* sp. BMCs (Fig. 3), totaling 22 genomes used for this pangenome analysis. For the *Cobetia* sp. pangenome, 10 genomes were used in addition to the *Cobetia* sp. BMC 6, totaling 11 genomes. The selection of these 10 genomes was based on the results of the taxonomic analysis of the ANI (Table S2) and on the clade formed with BMC 6 in the phylogenomic tree (Fig. 1). Finally, for the *Halomonas* sp. pangenome, 3 genomes were used (*H. taeanensis* BH539, *H. taeanensis* USBA-857, and *Halomonas* sp. YLGW01) in addition to the *Halomonas* sp. BMC 7, totaling 4 genomes. The selection of these 3 genomes was also based on the results provided by the taxonomic analysis of the ANI (Table S2) and on the clade formed with BMC 7 in the phylogenomic tree (Fig. 2). All genomes used for comparison with the BMC genomes were obtained from the PATRIC database (112). The pangenome figures were created using the Anvi'o visualization platform (120).

The categorization of genes used the results provided by the pangenomic analysis. Singleton genes from each of the BMCs were used for the purpose of analyzing which types of genes were unique to each BMC and to verify their potential to promote any benefit to the host. We define singleton genes as genes that are present in only one strain of the BMC or genes that were shared only among the BMCs but that do not appear in the genomes of other members of the genus included in the analyses.

**Data availability.** The complete genome sequencing data, including raw sequence reads, genome assemblies, and annotations of the BMC strains (BMC 1, BMC 2, BMC 3, BMC 4, BMC 5, BMC 6, and BMC 7) used in this study were submitted to NCBI, GenBank under the BioProject accession PRJNA638634 and BioSample accession numbers SAMN15198640 to SAMN15198646. The genomes used for comparison are available in Table S3.

## SUPPLEMENTAL MATERIAL

Supplemental material is available online only.

**FIG S1**, TIF file, 1.1 MB.

**TABLE S1**, DOCX file, 0.01 MB.

**TABLE S2**, DOCX file, 0.01 MB.

**TABLE S3**, DOCX file, 0.01 MB.

**TABLE S4**, DOCX file, 0.01 MB.

**TABLE S5**, DOCX file, 0.01 MB.

**TABLE S6**, DOCX file, 0.01 MB.

## ACKNOWLEDGMENTS

This work was supported by grants FCC/1/1973-51-01, REI/1/4984-01, and BAS/1/1095-01-01 from King Abdullah University of Science and Technology (R.S.P.).

P.M.R. and R.S.P. designed the study and wrote the paper. J.S. and U.R. performed the sequencing of the genomes. P.M.R., P.M.C., and J.G.R. carried out the bioinformatics analysis. P.M.R., P.M.C., R.S.P. and T.K.-C. analyzed and interpreted the data. P.M.R. and J.G.R. created the figures. R.S.P. funded the project. All authors read, revised, and approved the final manuscript.

We declare no conflict of interest.

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
