## [Reviewer comments · mSystems]

Exploring the potential molecular mechanisms of interactions between a probiotic consortium and its coral host

Phillipe Rosado, Pedro Cardoso, Joao Rosado, Junia Schultz, Ulisses Nunes da Rocha, Tina Keller-Costa, and Raquel Peixoto

Corresponding Author(s): Raquel Peixoto, King Abdullah University of Science and Technology

Review Timeline:

Submission Date:	September 19, 2022
Editorial Decision:	November 16, 2022
Revision Received:	December 5, 2022
Editorial Decision:	January 3, 2023
Revision Received:	January 3, 2023
Accepted:	January 4, 2023

Editor: Jeff Bowman

Reviewer(s): The reviewers have opted to remain anonymous.

Transaction Report:

DOI: <https://doi.org/10.1128/msystems.00921-22>

November 16, 2022

Prof. Raquel Peixoto
King Abdullah University of Science and Technology
Thuwal
Saudi Arabia

Re: mSystems00921-22 (Exploring the molecular mechanisms of interactions between a probiotic consortium and its coral host)

Dear Prof. Raquel Peixoto:

Thank you for submitting your manuscript to mSystems. We have completed our review and anticipate being able to accept the work after some minor but significant revisions. Please pay close attention to the reviewer comments and address each one in your response. In particular, both reviewers noted the lack of a control in the study design. Please be sure to address this and any caveats in your revised manuscript.

Preparing Revision Guidelines

Sincerely,

Jeff Bowman

Editor, mSystems

Journals Department
Reviewer comments:

Reviewer #1 (Comments for the Author):

General comments:

The manuscript "Exploring the molecular mechanisms of interactions between a probiotic consortium and its coral host" presents a survey of the genomes of hypothesized "beneficial microbes for corals" (BMCs) and highlights potential key genes and mechanisms through which these BMCs may support coral health during thermal stress. The data and take-aways are wonderfully presented and the manuscript is generally well-written; I also believe that this manuscript will be of great interest to those studying coral probiotics, as well as the coral reef community in general. I do have some concerns about how well the approach used in the manuscript is suited to answer the question that underpins this work.

Specifically, the sequenced genomes were derived from BMCs used in previous work from the same authors. In that publication, bacterial strains were screened for potential probiotics activity (antibiotics production, ROS scavenging, etc.); subsequent inoculations of heat-stressed corals with these bacterial strains then resulted in reduced bleaching rates. This was a very interesting and important finding and certainly merits further investigation into the mechanisms underpinning these results. Yet, the original experiment lacked strong experimental controls consisting of, for example, inoculations with bacterial strains that did not pass the screening. Including such controls may have aided the identification of specific functions that underpin BMC activity. Additionally, inoculations with heat-killed bacteria could have been used to control for the addition of important nutrients that may alleviate coral bleaching by themselves (without any actual bacterial activity). Due to the lack of these controls, surveying the genomes of BMCs from this experiment for genes that might alleviate bleaching might be viewed, to some extent, as circular reasoning. If the authors had included bacterial inoculations that were not expected to alleviate bleaching in the original experiment, these strains would have served as excellent controls to compare gene compositions against.

Having shared these concerns, I still think that the findings presented in the current manuscript will contribute to advancing the fast-growing field of coral probiotics. Rosado and colleagues also elegantly addressed some of my concerns in the current manuscript by providing comparisons against genomes of related bacterial strains that have not been demonstrated to alleviate bleaching. Still, I urge the authors to be cautious with their conclusions and to discuss some of these caveats in the manuscript. Overall, I think that this is a well-conducted study that will be of great interest to the scientific community

-

Other general comments:

Although the analyses are well presented and informative, I am missing some type of overview of how common the identified genes of interest are in bacterial strains generally, or (coral-associated) microbiomes specifically. I think it would bolster the manuscript to include a survey of published metagenomes and-maybe even more interesting-metatranscriptomes for the genes of interest. Perhaps some of these genes are expressed at higher rates in corals with higher stress resistance, for example. This would further alleviate some of the concerns I outlined in the paragraph above.

Throughout the text, there are some sections and sentences that seem a little out of place or it seems like there are missing/incorrect words. I would make sure to proofread the manuscript closely a few more times before the next (re-)submission.

Some specific comments:

Title:

No actual interactions were quantified in this manuscript so I would advise revising to "potential molecular mechanisms".

Abstract:

Line 17 - This is somewhat subjective, but I would urge steering away from using "proven" in the very first sentence since any finding can technically be disproven; I suggest using "shown" or "demonstrated".

Line 18 - Reconsider using the word "medicine", maybe just stick with probiotics. Also, is it really "efficient" (e.g., cost efficient, time efficient)? I think "effective" is more accurate.

Introduction:

Lines 35-37-doesn't follow from previous sentences, I would rephrase to something like "higher diversity of microbes has been linked to increased access to other food types" or just cut entirely.

Lines 62-65-This needs to be re-written to be clearer and offer a little more detail. It might be helpful to the reader to include a table or figure that includes some of the strains used, the findings from the initial screening, and the treatments used in the initial experiment.

Lines 65-67- again, replace proved throughout the text with something else, e.g., demonstrated or shown.

Methods:

Lines 206-207-Reads like something is missing in this sentence.

Results:

No major concerns here

Discussion:

Line 386-Replace "in between" with just "between".

Line 400-401-the fatty acid and quinone analyses seem very specific; I suggest either elaborating or cutting.

Line 417-define GSS

Line 420-428-This section on bleaching reads very introduction-y, covers a lot of different concepts, and feels like it's in a weird location. I would suggest streamlining a little bit; perhaps change to flow more logically from the introduction of glutathione peroxidase using a sentence like "H₂O₂, among other ROS, are produced in coral holobionts by during periods of high temperatures and irradiance, causing damage to host and symbiont cells. The production of ROS is directly linked to coral bleaching; ROS-scavenging BMC strains may therefore alleviate bleaching signs in stressed coral holobionts and should therefore be targeted for BMC selection."

Lines 439: This is not the first time siderophores are mentioned but it's the first time it's explained what siderophores do. Consider adding the explanation to earlier text.

Line 568: The use of the word "paramount" makes it read like this is the most important mechanism discussed thus far. Not sure if that is on purpose. Consider toning down.

Figures:

Figures 2-3: clarify what the green shaded area means. It is labeled "BMC #" but it's unclear what that means. Are those within 95% similarity for example?

Figure 7: This is a great summary figure, but the text in the panels is inconsistent and it might be good to have another look at what to include here. For example, in some panels it states directly how certain genes/functions may support coral health, but this information lacks in others (e.g., panels 2, 4, 5, 7). It would be nice to include that information in each panel and to be a little more specific where possible. For example, the text in panel 5 might be more informative like this: "Protect against ROS; may mitigate cellular damage in heat-stressed corals". The more specific, the better.

Add that these are "singleton" genes in the figure text (it now just says "unique" instead).

Panel 5: Projection-Should this say protection? If so, the "the" here is superfluous.

Panel 6: Capitalize F?

Reviewer #2 (Comments for the Author):

Overview Summary:

The authors use this manuscript to dive deeper into a previous paper that was published in 2019 that experimentally tested whether a consortium of potentially beneficial micro-organisms can be used as probiotics for corals undergoing stress. This manuscript takes the identified, cultured strains used in this consortium and sequences their genomes to examine the taxonomic identification and functional profiles of their genes, focusing particularly on those genes that were unique to each isolated strain. The authors conclude that these taxa have functions that benefit the coral host under stress.

Comments:

Both the writing and the *in silico* analysis for these isolated bacterial genomes was very well-executed. However, I am concerned about the context in which these data are presented and the conclusions that the authors make. This manuscript is a follow-on to a previous paper published by some of the authors (Rosado et al. 2019) and is contextualized as further exploration of the functions that benefitted the coral host as per the results of this previous paper.

The first big picture caveat I see is that the original paper (Rosado et al. 2019) did not control for the addition of a microbial inoculant to the corals (e.g., a heat-killed consortium of "BMCs"), which puts into question whether these identified micro-organisms that are explored *in silico* in the present manuscript were beneficial via their cellular functions or simply providing nutrition to the coral host. Although they discuss the potential for *Cobetia*, in particular, as possibly providing function over nutrition, it remains untested and inconclusive for this taxon and the other members of the consortium. While it is not necessarily pertinent to the bioinformatic analyses conducted in the present manuscript, I feel this issue is pertinent to how the results are contextualized - do the functions that were presented here matter to corals if they are simply eating the bacteria rather than

establishing a symbiosis? I feel some discussion of this either in the introduction and/or discussion of the manuscript is warranted.

Second, the aim of the paper according to the authors is to use this genome-exploration framework to aid in candidate selection for probiotic work (see lines 77-80). While I agree that genomic characterization is key to helping select potentially probiotic organisms by understanding what functions they may harbour, I think it is an oversight by the authors not to address the difference between functional potential and expression, particularly during stress events. Stress events (heat, salinity, pathogens, etc.) do not only impact the coral host cell function, but can also impact algal and bacterial cell function, causing expression that absolutely may be beneficial to the host, but could also not be. Perhaps in the introduction the authors need to clarify and expand their reasoning as to why a simple genome exploration study can provide useful information on how microbes confer stress resistance/resilience to corals, particularly without reference to their transcriptomes in vivo and under stress.

Lastly, I do appreciate the approach to looking at the genes that are unique to the 7 identified BMCs, however the authors jump quickly to the conclusion that these unique genes are beneficial to corals. Yet, it is not clear whether these genes are redundant in the larger coral microbiome. For instance, a unique oxidative stress response gene in an identified BMC does not necessarily mean that the coral microbiome does not already have existing oxidative stress response pathways. How does the framework presented here provide a mechanism for identifying that a unique gene (or set of genes) in a BMC is more beneficial to the coral host than genes for the same pathway that are present in other members of the coral microbiome?

As I mentioned above, I think the execution of the in silico analysis is quite well done (I had no comments or concerns with the methods of this work), and I particularly enjoyed exploring the pangemone visualizations. However, there are many assumptions about the benefits to coral host health that frame the manuscript, and I would like to see the authors better address the caveats to their concept throughout. Additionally, I think the authors need to re-consider or perhaps further synthesize how they might use this framework to identify putative beneficial bacteria for use in inoculant studies.

For specific comments, please see below by line number:

Lines 59-61: The original coral probiotic hypothesis needs to be cited here: Reshef et al. 2006 Environmental Microbiology

Line 285: Grammatical error - remove the word "other" or replace with "an additional"

Lines 364-365: Instead of stating "Figure 7 shows..." it might be better to reference this figure in the discussion as a summary figure. However, I would caution against using a summary figure such as this at all when the results cannot be connected to coral health without further transcriptomic or metabolic confirmation.

Lines 396-407: It should be added that a lack of comparative genomes also likely contributed to the low taxonomic resolution of both the *Cobetia* and the *Halomonas* BMCs.

Section starting on line 408: It is unclear at the start of this section that the genes discussed here are not unique to the 7 BMCs (This is not clarified until line 495). Without clarifying in text, it clouds the interpretation of these data to the reader.

Lines 427-428: The authors state that highly effective oxidative stress response genes should be targeted for selection of potential BMCs, but how do we know that these genes are a) highly effective and b) are expressed in vivo?

Lines 434-437: How would synthesis of cobalamin be beneficial during stress? Is the idea that it would help maintain the relationship between the coral host and the symbiont and reduce bleaching? How do we know that stress doesn't impact this cellular machinery in the bacteria?

Lines 446- 451: If we know that other bacterial taxa perform these functions in the microbiome, where does this place your identified BMCs?

Lines 459-460: I think "infer" is too strong a word here - perhaps use "hypothesize" instead?

Line 464: I suspect this should read something more along the lines of "...we detected genes whose functions are involved in the synthesis of glycine betaine" rather than you having actually detected the synthesis of the compound itself.

Lines 467-470: Does this reference discuss the conditions under which GB metabolism acts on coral physiology? E.g., under stress or neutral conditions?

Lines 483-486: Do this mean that the BMC consortium included a high abundance of strains that had GB metabolism genes or that you found a high abundance of GB metabolism genes in the consortium? Either way, how do you know that other members of the microbiome do not carry similar genes?

Lines 487-494: This needs to be a much stronger statement here about the caveats of the study - it is suggested here that "in

some cases" follow-up transcriptomics and metabolomics are needed. I would argue that in ALL cases follow-up transcriptomics and metabolomics are necessary. It is crucially important to understand what genes are expressed under neutral and stress conditions to identify whether they have any role in providing beneficial cellular functions to their coral host.

Line 500: "encode" should be "encoded"

Lines 508-510: It is misleading to state that genes of unknown functions may play a vital role in holobiont health without also mentioning that they may also have no benefit to coral health.

Lines 527-529: I think it should be stated here that this is likely the case because there were limited data for these pangenome analyses, particularly for Halomonas where there only 3 representative genomes. This caveat is particularly important to note because sulfur and nitrogen metabolism can be conducted by many different members of the microbiome across many different genes. If the point of a genomic analysis is to select for potentially beneficial bacteria, how do you argue that these identified BMCs provide more of a benefit in terms of nutrient cycling/metabolism than other microbiome members who perform the same or similar functions? It might be worth considering & discussing the role this framework plays in BMC candidate selection.

Response to reviewer and editor comments

Exploring the potential molecular mechanisms of interactions between a probiotic consortium and its coral host

Editor's Comments to Authors:

Thank you for submitting your manuscript to mSystems. We have completed our review and anticipate being able to accept the work after some minor but significant revisions. Please pay close attention to the reviewer comments and address each one in your response. In particular, both reviewers noted the lack of a control in the study design. Please be sure to address this and any caveats in your revised manuscript.

Thank you for the privilege of reviewing your work.

Dear editor,

We attach the revised version of the manuscript "Exploring the potential molecular mechanisms of interactions between a probiotic consortium and its coral host". We were delighted to see the interest expressed in this work. We would like to thank you and the reviewers for the detailed reviews, constructive suggestions, and critical feedback provided. We have carefully addressed all comments, as detailed below. "
. We also would like to thank you and the reviewers for the detailed review and constructive suggestions provided.

Reviewer comments:

Reviewer #1 (Comments for the Author):

General comments:

The manuscript "Exploring the molecular mechanisms of interactions between a probiotic consortium and its coral host" presents a survey of the genomes of hypothesized "beneficial microbes for corals" (BMCs) and highlights potential key genes and mechanisms through which these BMCs may support coral health during thermal stress. The data and take-aways are wonderfully presented and the manuscript is generally well-written; I also believe that this manuscript will be of great interest to those studying coral probiotics, as well as the coral reef community in general. I do have some concerns about how well the approach used in the manuscript is suited to answer the question that underpins this work.

Specifically, the sequenced genomes were derived from BMCs used in previous work from the same authors. In that publication, bacterial strains were screened for potential probiotics activity (antibiotics production, ROS scavenging, etc.); subsequent

inoculations of heat-stressed corals with these bacterial strains then resulted in reduced bleaching rates. This was a very interesting and important finding and certainly merits further investigation into the mechanisms underpinning these results. Yet, the original experiment lacked strong experimental controls consisting of, for example, inoculations with bacterial strains that did not pass the screening. Including such controls may have aided the identification of specific functions that underpin BMC activity. Additionally, inoculations with heat-killed bacteria could have been used to control for the addition of important nutrients that may alleviate coral bleaching by themselves (without any actual bacterial activity). Due to the lack of these controls, surveying the genomes of BMCs from this experiment for genes that might alleviate bleaching might be viewed, to some extent, as circular reasoning. If the authors had included bacterial inoculations that were not expected to alleviate bleaching in the original experiment, these strains would have served as excellent controls to compare gene compositions against.

Having shared these concerns, I still think that the findings presented in the current manuscript will contribute to advancing the fast-growing field of coral probiotics. Rosado and colleagues also elegantly addressed some of my concerns in the current manuscript by providing comparisons against genomes of related bacterial strains that have not been demonstrated to alleviate bleaching. Still, I urge the authors to be cautious with their conclusions and to discuss some of these caveats in the manuscript. Overall, I think that this is a well-conducted study that will be of great interest to the scientific community.

A: We welcome your comments and appreciation of how this genomic analysis was performed and also its importance for studies on coral probiotics.

Regarding your concerns about the different controls that we could have used in the first experiment described in the paper published in 2019, we agree that using a consortium with bacteria that did not pass the screening would be an excellent way to compare probiotic effects and include unknown mechanisms. However, it is important to consider that the vast majority of bacteria that do not pass the screening are potential coral pathogens. Furthermore, this was the first experiment carried out with BMC inoculation, in which the bacterial characteristics we were looking for were based on characteristics described in the literature at the time that could be putatively beneficial for the coral holobiont. In fact, we cannot exclude any bacteria as probiotics without testing them, as there might be beneficial mechanisms we are not aware of. We have therefore followed the plan of selecting putative ones, as hypothesized in the literature (Peixoto et al., 2017 and 2021) to validate them (or not) and, at the same time, take advantage of the manipulative experiments and track down other beneficial mechanisms using several omics (Santoro et al., 2021). As we run new experiments and enrich our database with information on the use of BMC in corals, we continue to add new controls and new analyses such as metatranscriptomics, metagenomics, metabolomics, and so on. This type of comparison with cells that did not pass the screening forms part of some BMC

experiments that we are planning to carry out once we gather more information, which also aim to validate if it is possible to exclude some of them in our screening. Regarding the inoculations with heat-killed bacteria, we believe their use depends on the research question, as the use of dead cells (also called 'postbiotics') is not suitable as a negative control because their use may trigger specific responses from the host and therefore incorporate a confounding factor. Negative controls allow the validation of treatment effect by ruling out or minimizing the effects of any confounding variables and, therefore, should be as inert as possible. A placebo typically includes the administration/inoculation of the same inert carrier and/or procedure used in the main treatment. We have reviewed this in a paper currently in review in Trends in Microbiology, as it is a common point of discussion. We conclude that postbiotics can be tested as an alternative microbial treatment that can help describe the mechanisms of action of probiotics and postbiotics, but are not a reliable negative control. Hence, a placebo does not add potential confounding variables and tests whether any other factor, except the actual probiotic, may exert the observed/ascribed effects. As this first BMC inoculation experiment was only intended to test whether the inoculation of beneficial bacteria for the coral holobiont (based on the literature at the time) would have any beneficial effect on the coral in view of the increase in temperature and the pathogen challenge, we believe that the use of heat-killed bacteria would not be applicable in this specific experiment. We appreciate your comments and have added this explanation and caveats in the new version of the text, as follows: "Although this study did not aim to show that BMCs have certain characteristics that are not present in other members of the coral microbiome, our approach may facilitate the detection of microorganisms harboring these theoretically vital roles to the host health, which could therefore be tested as BMCs. The best way to validate the use of a BMC consortium as a means to mitigate damage caused by different stressors is to compare any health improvement with the use of a placebo treatment. Alternative microbial therapies, such as the use of heat-killed bacteria (also called postbiotics) can also help to understand the protective mechanisms provided by microbes (dead or alive). The use of microbes that were not selected by current screening methods can also generate data on unknown beneficial mechanisms."

Other general comments:

Although the analyses are well presented and informative, I am missing some type of overview of how common the identified genes of interest are in bacterial strains generally, or (coral-associated) microbiomes specifically. I think it would bolster the manuscript to include a survey of published metagenomes and maybe even more interesting-metatranscriptomes for the genes of interest. Perhaps some of these genes are expressed at higher rates in corals with higher stress resistance, for example. This would further alleviate some of the concerns I outlined in the paragraph above.

A: Thank you for pointing this out and for the suggestion. We agree and have added a new paragraph in the discussion as follows: “In summary, our results indicate several possible theoretical ways in which BMCs could have acted to help corals during periods of stress (7). Although, as highlighted above, most of these characteristics are still theoretical, some of the putative beneficial genes have already been validated or shown to have differential expression during heat stress events in corals. For instance, metatranscriptome analyses have shown the differential expression of several genes in response to thermal stress in scleractinian corals, such as *Musismilia hispida*, *Siderastrea radians*, *Orbicella faveolata*, and *Pseudodiploria clivosa* (10, 119). The results reveal that several genes previously suggested to be potentially beneficial to corals (and also detected in our genomes) were upregulated during stress, such as peroxidase genes related to protection against ROS (e.g., *S. radians* and *O. faveolata*) and vitamin B biosynthesis (e.g., *P. clivosa*). Genes related to sulfur and nitrogen metabolism were also upregulated, with a higher expression in *S. Radians*. This differential expression was mainly due to shifts in the microbiome, when compared to the host and the endosymbiotic alga, suggesting that the microbiome is primarily responsible for the upregulation of these genes in *S. radians*.”

Throughout the text, there are some sections and sentences that seem a little out of place or it seems like there are missing/incorrect words. I would make sure to proofread the manuscript closely a few more times before the next (re-)submission.

Some specific comments:

Title:

No actual interactions were quantified in this manuscript so I would advise revising to "potential molecular mechanisms".

A: We thank the reviewer for the suggestion, and we have updated the title.

Abstract:

Line 17 - This is somewhat subjective, but I would urge steering away from using "proven" in the very first sentence since any finding can technically be disproven; I suggest using "shown" or "demonstrated".

A: Thank you for pointing this out. We agree and we have changed the word.

Line 18 - Reconsider using the word "medicine", maybe just stick with probiotics. Also, is it really "efficient" (e.g., cost efficient, time efficient)? I think "effective" is more accurate.

A: Thank you for the suggestion. Both words have been changed.

Introduction:

Lines 35-37-doesn't follow from previous sentences, I would rephrase to something like "higher diversity of microbes has been linked to increased access to other food types" or just cut entirely.

A: Good point. We agree and have changed the original text in accordance with your suggestion.

Lines 62-65-This needs to be re-written to be clearer and offer a little more detail. It might be helpful to the reader to include a table or figure that includes some of the strains used, the findings from the initial screening, and the treatments used in the initial experiment.

A: Thank you for pointing this out. We have added more information about the BMC inoculation experiment in this paragraph, as well as a table summarizing the information as suggested.

Lines 65-67- again, replace proved throughout the text with something else, e.g., demonstrated or shown.

A: Well noted. The word proved has been changed throughout the manuscript.”.

Methods:

Lines 206-207-Reads like something is missing in this sentence.

A: Thank you for pointing out that this sentence was unclear. Here, we wanted to justify the purpose of using and analyzing singleton genes. We changed the sentence slightly, as follows, to improve readability:

“Singleton genes from each of the BMCS were used for the purpose of analyzing which types of genes were unique to each BMC and to verify their potential to promote any benefit to the host. We define singleton genes as genes that are present in only one strain of the BMC or genes that were shared only among the BMCs, but that do not appear in the genomes of other members of the genus included in the analyses”

Results:

No major concerns here

Discussion:

Line 386-Replace "in between" with just "between".

A: Changed.

Line 400-401-the fatty acid and quinone analyses seem very specific; I suggest either elaborating or cutting.

A: Thank you for this observation. We have edited the sentence, as follows: "*Cobetia* sp. BMC 6 may represent a new species within the genus *Cobetia*, but further analyses, such as phenotypic and biochemical tests related to the genus and API tests, are necessary to determine this (68)."

Line 417-define GSS

GSS is the name of the gene responsible for coding the glutathione synthetase protein. We have changed the sentence to better explain the role of this gene, as follows: "Another important gene associated with the antioxidant role found in all BMCs is the gene named GSS which encodes the protein glutathione synthetase, and is related to the glutathione biosynthesis pathway"

Line 420-428-This section on bleaching reads very introduction-y, covers a lot of different concepts, and feels like it's in a weird location. I would suggest streamlining a little bit; perhaps change to flow more logically from the introduction of glutathione peroxidase using a sentence like "H₂O₂, among other ROS, are produced in coral holobionts by during periods of high temperatures and irradiance, causing damage to host and symbiont cells. The production of ROS is directly linked to coral bleaching; ROS-scavenging BMC strains may therefore alleviate bleaching signs in stressed coral holobionts and should therefore be targeted for BMC selection."

A: Thank you for the suggestion, which we have followed since we agree that this part contained a lot of introductory material. The section now reads: "...glutathione can be used by the protein glutathione peroxidase, also present in all BMCs, to neutralize reactive oxygen species (ROS), such as H₂O₂ (71). H₂O₂, among other ROS, are produced in coral holobionts during periods of high temperatures and irradiance, causing damage to host and symbiont cells (72-74). The production of ROS is directly linked to coral bleaching (75,76). ROS-scavenging BMC strains have been hypothesized to alleviate bleaching signs in stressed coral holobionts and should therefore be targeted for BMC selection."

Lines 439: This is not the first time siderophores are mentioned but it's the first time it's explained what siderophores do. Consider adding the explanation to earlier text.

A: Thank you for this observation. We have now added the function of siderophores when they are first mentioned, as follows: "...siderophore production (i.e., production of iron-chelating compounds)..." We then explore it in more details at: "Siderophores are organic compounds that act to capture iron from the environment and their production has also been suggested as a BMC trait (55). Iron is essential for several physiological processes in corals and microalgae, including photosynthesis, respiration, and nitrogen fixation (79); yet, bioavailable iron concentrations in most of the global oceans are too

low to support growth of microalgae (80). Several marine bacteria produce siderophores, which bind and concentrate iron in bioavailable forms, allowing the absorption of this limiting micronutrient by phytoplankton. For instance, the production of siderophores by the gammaproteobacterium *Marinobacter* promotes the growth of its dinoflagellate partner *Scrippsiella trochoidea* (81). *Marinobacter* strains are also a component of the “core microbiome” shared between different cultures of Symbiodiniaceae (82), which likely rely on bacterial partners to meet their bioavailable iron needs.”

Line 568: The use of the word "paramount" makes it read like this is the most important mechanism discussed thus far. Not sure if that is on purpose. Consider toning down.

A: We agree that the use of “paramount” gives great emphasis to this mechanism. So, we have revised the sentence as follows: “Therefore, any mechanism that reduces the excess of intracellular NO in the holobiont seems to be important for the health of corals.”

Figures:

Figures 2-3: clarify what the green shaded area means. It is labeled "BMC #" but it's unclear what that means. Are those within 95% similarity for example?

A: This is a very good point. The area shaded in green represents the lineages that formed a monophyletic group with BMC6 and 7. This information was added to the legend to facilitate the understanding of the figure as follows: “Fig. 2: Phylogenomic inference of publicly available genomes from 24 *Cobetia* strains and the genomes of BMC6 strain (in red), totaling 25 genomes. The green shaded area represents the strains that form a monophyletic clade with BMC6. The tree was assembled from the comparison of 1000 proteins through the codon tree method of the PATRIC platform...” and **FIG 3** Phylogenomic inference of publicly available genomes from 39 *Halomonas* strains and the genomes of BMC7 strain (in red), totaling 40 genomes. The green shaded area represents the strains that form a monophyletic clade with BMC7...”

Figure 7: This is a great summary figure, but the text in the panels is inconsistent and it might be good to have another look at what to include here. For example, in some panels it states directly how certain genes/functions may support coral health, but this information lacks in others (e.g., panels 2, 4, 5, 7). It would be nice to include that information in each panel and to be a little more specific where possible. For example, the text in panel 5 might be more informative like this: "Protect against ROS; may mitigate cellular damage in heat-stressed corals". The more specific, the better. Add that these are "singleton" genes in the figure text (it now just says "unique" instead).

Panel 5: Projection-Should this say protection? If so, the "the" here is superfluous.

Panel 6: Capitalize F?

A: We have incorporated all of these excellent suggestions as follows:

“Panel 2: B-complex vitamins are cofactors for several proteins within the holobiont.

Panel 4: Nitrogen is essential for the synthesis of proteins, nucleic acids, and other components of cells.

Panel 5: May mitigate cellular damage in heat-stressed corals.

Panel 7: The product of the activity of this gene can act as an antioxidant and is involved in the production of some amino acids.”

Reviewer #2 (Comments for the Author):

Overview Summary:

The authors use this manuscript to dive deeper into a previous paper that was published in 2019 that experimentally tested whether a consortium of potentially beneficial micro-organisms can be used as probiotics for corals undergoing stress. This manuscript takes the identified, cultured strains used in this consortium and sequences their genomes to examine the taxonomic identification and functional profiles of their genes, focusing particularly on those genes that were unique to each isolated strain. The authors conclude that these taxa have functions that benefit the coral host under stress.

Comments:

Both the writing and the in silico analysis for these isolated bacterial genomes was very well-executed. However, I am concerned about the context in which these data are presented and the conclusions that the authors make. This manuscript is a follow-on to a previous paper published by some of the authors (Rosado et al. 2019) and is contextualized as further exploration of the functions that benefitted the coral host as per the results of this previous paper.

The first big picture caveat I see is that the original paper (Rosado et al. 2019) did not control for the addition of a microbial inoculant to the corals (e.g., a heat-killed consortium of "BMCs"), which puts into question whether these identified micro-organisms that are explored in silico in the present manuscript were beneficial via their cellular functions or simply providing nutrition to the coral host. Although they discuss the potential for *Cobetia*, in particular, as possibly providing function over nutrition, it remains untested and inconclusive for this taxon and the other members of the consortium. While it is not necessarily pertinent to the bioinformatic analyses conducted in the present manuscript, I feel this issue is pertinent to how the results are contextualized - do the functions that were presented here matter to corals if they are simply eating the bacteria rather than establishing a symbiosis? I feel some discussion of this either in the introduction and/or discussion of the manuscript is warranted.

A: Firstly, we would like to thank you for your comments, which have significantly contributed to the improvement of our manuscript.

As mentioned in our response to Reviewer 1, this is a common topic of discussion. The use of a control using heat-killed bacteria in the experiment of the previous paper is frequently raised and, not surprisingly, both reviewers commented on this issue. The point is that the use of this control in experiments would introduce bias. As we mentioned above, the use of heat-killed bacteria as a negative control may trigger specific responses from the host and therefore incorporate a confounding factor in the experiment. We have a “good practice” paper in review now in Trends in Microbiology in which we deeply discuss this, using a systematic review of ~ 500 papers on probiotics to support the use of placebo as the best control for any probiotic study (across different hosts). As this work published in 2019 was the first to test a putative probiotic consortium, there were no previous studies showing the use of probiotics in corals at that time. We, therefore, asked: i, Is it possible, in some way, to manipulate the microbiome by adding strains that passed through our screening? ii, Would this BMC consortium bring any benefit to the coral in terms of protection against thermal stress and pathogen inoculation? For this, a very large experiment was carried out with 2 different temperatures and 8 treatments in triplicate. We found a significant difference in the treatments with BMC inoculation and, based on this, our next works should be precisely focused on understanding the mechanisms responsible for corals becoming healthier or suffering less damage from both stresses when inoculated with BMC. In another approach (Santoro et al., 2021), we tracked the mechanisms being up- and down-regulated to validate some of the hypothesized mechanisms, such as DMSP degradation. But this is a field in its infancy, and the more we manipulate and track the results the more we will confirm some of the mechanisms and discover others. The use of a genome-based screening can accelerate such tests. The fact that we were able to find sequences from BMC6, *Cobetia*, in the amplicon metagenomic analyses that we performed on corals was just one more indication that we were able to manipulate the microbiome (which was one of the questions in our previous work). Therefore, one of the aims of this current work with the analysis of BMC genomes is to describe and discuss possible reasons why BMC inoculation worked, in addition to being a framework for selecting new BMCs. We appreciate your comments and have added this explanation and caveats in the new version of the text, as follows: “Although this study did not aim to show that BMCs have certain characteristics that are not present in other members of the coral microbiome, our approach may facilitate the detection of microorganisms harboring these theoretically vital roles to the host health, which could therefore be tested as BMCs. The best way to validate the use of a BMC consortium as a means to mitigate damage caused by different stressors is to compare any health

improvement with the use of a placebo treatment. Alternative microbial therapies, such as the use of heat-killed bacteria (also called postbiotics) can also help to understand the protective mechanisms provided by microbes (dead or alive). The use of microbes that were not selected by current screening methods can also generate data on unknown beneficial mechanisms.”

Second, the aim of the paper according to the authors is to use this genome-exploration framework to aid in candidate selection for probiotic work (see lines 77-80). While I agree that genomic characterization is key to helping select potentially probiotic organisms by understanding what functions they may harbour, I think it is an oversight by the authors not to address the difference between functional potential and expression, particularly during stress events. Stress events (heat, salinity, pathogens, etc.) do not only impact the coral host cell function, but can also impact algal and bacterial cell function, causing expression that absolutely may be beneficial to the host, but could also not be. Perhaps in the introduction the authors need to clarify and expand their reasoning as to why a simple genome exploration study can provide useful information on how microbes confer stress resistance/resilience to corals, particularly without reference to their transcriptomes in vivo and under stress.

A: Thank you for pointing this out and for the suggestion. We added a new paragraph in the introduction showing the nuances of a study based mainly on genomic analyses and the need for analyses that demonstrate the expression of these genes in future works, as follows: “It is important to emphasize that the beneficial characteristics for corals analyzed in this work are mainly theoretically based on the literature, which require validation by, for example, combined physiological and molecular (such as metatranscriptomics and other omics) monitoring. Genomic surveys would be an important addition as a first step towards setting up a probiotic consortium for corals that will then need to be tested and validated in laboratory experiments or field trials.”

It is important to point out, as previously mentioned, that we also added a paragraph at the end of the discussion exemplifying metatranscriptomic studies that demonstrated the increased expression of several genes that we discuss in this paper with potential beneficial characteristics for corals.

Lastly, I do appreciate the approach to looking at the genes that are unique to the 7 identified BMCs, however, the authors jump quickly to the conclusion that these unique genes are beneficial to corals. Yet, it is not clear whether these genes are redundant in the larger coral microbiome. For instance, a unique oxidative stress response gene in an identified BMC does not necessarily mean that the coral microbiome does not already have existing oxidative stress response pathways. How does the framework

presented here provide a mechanism for identifying that a unique gene (or set of genes) in a BMC is more beneficial to the coral host than genes for the same pathway that are present in other members of the coral microbiome?

A: Thank you very much for the good question. Our goal was not to only look for traits or genes that are uniquely present in BMC genomes. All these characteristics that we believe to be beneficial were chosen based on findings described in the literature, so one of our aims was to seek these characteristics in our BMCs and, through the inoculation of the consortium, increase the number of bacteria capable of bringing such beneficial roles to the holobiont, thus enhancing certain functional roles within the holobiont. The genomic comparison of our BMC with other strains through pangenome analysis was just a way to show that our BMCs have several genes which, in theory, are good for the holobiont, and that increasing the number of microorganisms that have these functional characteristics could help corals in the face of some stress, and therefore also offer a roadmap for other authors to track such traits in their strains. We have highlighted this as detailed in our answer above.

As I mentioned above, I think the execution of the *in silico* analysis is quite well done (I had no comments or concerns with the methods of this work), and I particularly enjoyed exploring the pangenome visualizations. However, there are many assumptions about the benefits to coral host health that frame the manuscript, and I would like to see the authors better address the caveats to their concept throughout. Additionally, I think the authors need to re-consider or perhaps further synthesize how they might use this framework to identify putative beneficial bacteria for use in inoculant studies.

A: Thank you very much for your comments, and we are glad that you enjoyed the pangenome analyses. We believe we have already addressed caveats throughout the text, as can be seen in our responses above. For instance, we have added paragraphs explaining the need for metatranscriptomic (and other omics) analysis in all experiments with BMC, as well as exemplifying metatranscriptomic studies showing differential expression of putative beneficial genes for corals during heat stress. In addition, we also address the topic of using alternative microbial treatments in future experiments, such as bacteria that did not pass the screening and/or heat-killed bacteria, and reinforce the use of placebo as the preferred negative control.

We have also added a sentence at the end of the discussion that helps to explain the importance of using *in silico* analyses for BMC selection, as follows: "...the use of *in silico* screenings can simultaneously accelerate the selection of cultured BMCs (since through genome analysis it is possible to quickly identify a greater variety of genes when compared to PCR and biochemical/physiological assays) and improve our ability to culture alternative BMCs."

For specific comments, please see below by line number:

Lines 59-61: The original coral probiotic hypothesis needs to be cited here: Reshef et al. 2006 Environmental Microbiology

A: Good point. We have added the reference to the text as requested.

Line 285: Grammatical error - remove the word "other" or replace with "an additional"

A: Well noted, this error has been fixed.

Lines 364-365: Instead of stating "Figure 7 shows..." it might be better to reference this figure in the discussion as a summary figure. However, I would caution against using a summary figure such as this at all when the results cannot be connected to coral health without further transcriptomic or metabolic confirmation.

A: Thank you for the suggestion. We agree with you and now highlight in the legend that this is a **theoretical illustration** of the functional roles that BMCs might play within the holobiont, as follows: "Theoretical illustration of the benefits that singleton genes of BMCs might bring to the coral host and its endosymbiotic algae." This theoretical summary may be helpful for BMC screenings that can then validate their role (which we have also clarified as per the highlighted caveats along the text).

Lines 396-407: It should be added that a lack of comparative genomes also likely contributed to the low taxonomic resolution of both the *Cobetia* and the *Halomonas* BMCs.

A: Good point. We agree and have added it to the text as follows: "This BMC is possibly a new species of *Halomonas*, but further analyses to characterize the species are needed, as described for *Cobetia* sp. BMC 6. Based on this evidence, the lack of comparative genomes likely contributed to the low taxonomic resolution of both the *Cobetia* and the *Halomonas* BMCs."

Section starting on line 408: It is unclear at the start of this section that the genes discussed here are not unique to the 7 BMCs (This is not clarified until line 495). Without clarifying in text, it clouds the interpretation of these data to the reader.

A: Good catch. We have added a sentence throughout this section to clarify that these genes are not unique to the BMC strains, as follows: "Following the phylogenomic and taxonomic inference of the BMCs, we searched their genomes for genes encoding proteins that could be potentially beneficial for corals. These genes are not exclusive to BMCs, and can also be found in any non-BMC strains. In addition to genes encoding catalase and nitrite reductase enzymes identified via PCR and biochemical tests..."

Lines 427-428: The authors state that highly effective oxidative stress response genes

should be targeted for selection of potential BMCs, but how do we know that these genes are a) highly effective and b) are expressed in vivo?

A: We do not know whether these genes are highly effective; our statements are based on previous studies that demonstrated the effectiveness, often theoretical, of these genes in the detoxification of ROS. Regarding gene expression, many of these genes related to oxidative responses can be tested in simple laboratory tests, such as the catalase test that was performed during the screening of the BMCs in Rosado et al., 2019. All of these genes that we have suggested to be used as targets for screening new BMC strains are theoretical contexts based on findings in the literature. It may be that they do not act as we expect or even are not being expressed. These caveats were included in the text, as detailed above, and also as follows:

“The production of ROS is directly linked to coral bleaching (75,76). ROS-scavenging BMC strains have been hypothesized to alleviate bleaching signs in stressed coral holobionts and should therefore be targeted for BMC selection.”

Lines 434-437: How would synthesis of cobalamin be beneficial during stress? Is the idea that it would help maintain the relationship between the coral host and the symbiont and reduce bleaching? How do we know that stress doesn't impact this cellular machinery in the bacteria?

A: We believe that cobalamin may be important for the holobiont because its biosynthesis is related to several metabolic pathways, including some that might directly help corals during heat stress, such as DMSP antioxidant synthesis. In addition, Symbiodiniaceae is not capable of synthesizing cobalamin, so adding BMCs that may play this role would help to meet the needs of endosymbiotic algae. Again, this is theory that is described in the literature (e.g., Peixoto et al., 2017 and 2021, Matthews et al., 2020, Santoro et al., 2021); several more specific studies are needed to substantiate this theory. Some such studies are already being performed by our laboratory, as well as in other laboratories. We believe the caveats discussed above will be important to highlight that this is a roadmap for the screening of BMCs that will be then validated using manipulative studies followed by multi-omics and physiological analyses, as for example in Santoro et al., 2021.

Lines 446- 451: If we know that other bacterial taxa perform these functions in the microbiome, where does this place your identified BMCs?

A: One of the aims of using BMC is to increase (or retain, as some beneficial microbes are often replaced by pathogens during thermal stress events) the number of microorganisms that have certain functions that, based on the literature, may be beneficial to the holobiont. The fact that some bacteria have already been described because of their functions within the holobiont does not make the use of BMCs unfeasible, it even helps because it shows us that certain characteristics can in fact be beneficial. In this case we are trying to retain common, abundant, and native bacteria to

retain their beneficial function and, at the same time, occupy the niche to avoid pathogen overgrowth, although we agree that different bacterial groups can perform this function, and are testing some of them – which is also why broader screening processes may help increasing the identification of putative BMCs.

Lines 459-460: I think "infer" is too strong a word here - perhaps use "hypothesize" instead?

A: We agree. Fixed, thank you.

Line 464: I suspect this should read something more along the lines of ". We detected genes whose functions are involved in the synthesis of glycine betaine" rather than you having actually detected the synthesis of the compound itself.

A: Good point. We have changed the sentence.

Lines 467-470: Does this reference discuss the conditions under which GB metabolism acts on coral physiology? E.g., under stress or neutral conditions?

A: Thank you for pointing this out. The authors do not specifically discuss whether GB acts during stress conditions or neutral conditions. They do, however, list several potential roles that GB may have in the holobiont such as osmolytic properties, being the main nitrogen reserve in the holobiont, and a key role in carbon metabolism, mainly with regard to DNA methylation and protein synthesis. Furthermore, the authors also suggest that GB may have a thermo- and photoprotective role in the holobiont since it has this role in plants. Based on all these findings, the authors hypothesize a putative role for GB in stress resilience in coral metaorganisms. We have added a sentence to the discussion to make this clearer for readers, as follows: "...GB sequestration and storage to support metabolic activity, maintaining a proper osmotic environment while serving as a nitrogen reservoir to accommodate the growing needs of endosymbionts (86). It is important to note that these studies did not directly test the role of GB in heat-stressed corals, thus future research addressing this topic is required. For these reasons, in addition to the high abundance of GB-producing strains in our BMC consortium..."

Lines 483-486: Do this mean that the BMC consortium included a high abundance of strains that had GB metabolism genes or that you found a high abundance of GB metabolism genes in the consortium? Either way, how do you know that other members of the microbiome do not carry similar genes?

A: Our meaning is that the BMC consortium includes several strains that carry the genes for GB metabolism. We don't know if other members of the microbiome carry these genes. We are trying to understand why BMC inoculation helped corals under

stress conditions. Therefore, we are hypothesizing different roles that BMCs might have within the holobiont. The fact that other members of the holobiont have this role would not exclude the importance of BMCs in having it. We believe it is clear now, as follows: ““For these reasons, in addition to the high abundance of GB-producing strains in our BMC consortium, we also suggest GB production...”. And here: “In some cases, they also represent new BMC targets **that should be investigated in future research efforts** (such as the production of different ROS scavengers, GB, and ectoines). Furthermore, transcriptomic and/or metabolomic surveys will always be required in thermal stress experiments to confirm whether these BMC mechanisms are being expressed and correlated with thermal protection (10).”

Lines 487-494: This needs to be a much stronger statement here about the caveats of the study - it is suggested here that "in some cases" follow-up transcriptomics and metabolomics are needed. I would argue that in ALL cases follow-up transcriptomics and metabolomics are necessary. It is crucially important to understand what genes are expressed under neutral and stress conditions to identify whether they have any role in providing beneficial cellular functions to their coral host.

A: Thank you for pointing this out. We agree and changed the sentence to convey that transcriptome and metabolomics analyses are always important to confirm whether the BMC mechanisms are being expressed and correlated with thermal protection, as follows: “In some cases, they also represent new BMC targets that should be investigated in future research efforts (such as the production of different ROS scavengers, GB, and ectoines). Furthermore, transcriptomic and/or metabolomic surveys will always be required in thermal stress experiments to confirm whether these BMC mechanisms are being expressed and correlated with thermal protection (10).”

Line 500: "encode" should be "encoded"

A: Fixed.

Lines 508-510: It is misleading to state that genes of unknown functions may play a vital role in holobiont health without also mentioning that they may also have no benefit to coral health.

A: We agree. We have amended the sentence by mentioning that genes with unknown functions may also play no vital role in the holobiont.

Lines 527-529: I think it should be stated here that this is likely the case because there were limited data for these pangenome analyses, particularly for Halomonas where there only 3 representative genomes. This caveat is particularly important to note because sulfur and nitrogen metabolism can be conducted by many different members of the microbiome across many different genes. If the point of a genomic analysis is to

select for potentially beneficial bacteria, how do you argue that these identified BMCs provide more of a benefit in terms of nutrient cycling/metabolism than other microbiome members who perform the same or similar functions? It might be worth considering & discussing the role this framework plays in BMC candidate selection.

A: Thank you for pointing this out. We added this limitation bias in the data from the pangenome analyses in the revised text. In this work, we are showing several possible theoretical ways in which BMCs could have acted to help corals during periods of stress. Our goal is not necessarily to show that our BMCs have certain characteristics that no other member of the microbiome has. In fact, the intention is to accelerate the screening and therefore increase the number of microorganisms identified as having these theoretically vital roles in the host health and that could, therefore, be tested as BMCs. This information is now presented in the text as follows: “The protein siroheme synthase is encoded by the *cysG* gene and was only present in the genomes of *Cobetia* sp. BMC 6 and *Halomonas* sp. BMC 7, being absent in the other genomes used in the pangenome analyses of both genera. This may have occurred because there were limited data for these pangenome analyses, particularly for the *Halomonas* genus, for which there were only three representative genomes.”

January 3, 2023

Prof. Raquel Peixoto
King Abdullah University of Science and Technology
Thuwal
Saudi Arabia

Re: mSystems00921-22R1 (Exploring the potential molecular mechanisms of interactions between a probiotic consortium and its coral host)

Dear Prof. Raquel Peixoto:

Thank you for submitting your manuscript to mSystems. We have completed our review and I am pleased to inform you that, in principle, we expect to accept it for publication in mSystems. However, acceptance will not be final until you have adequately addressed the reviewer comments.

Preparing Revision Guidelines

Sincerely,

Thulani Makhwanyane

Editor, mSystems

Journals Department
American Society for Microbiology
1752 N St., NW

Reviewer comments:

Reviewer #1 (Comments for the Author):

Overall, nice changes. The goal and limitations of the study are much clearer now.

Some minor points on the revisions:

Lines 35-37 it's still a little unclear how this relates to the previous sentences. I would just cut it or streamline it and make it part of the previous sentence.

Lines 84-90: While I'm happy to see this inclusion, this reads slightly awkwardly because it goes back and forth between limitations and the goal of the current paper. It is also unclear what exactly the last sentence adds that's not already included earlier in the paragraph. Consider trimming to something like this:

"The purpose of this work is to perform an in silico analysis of the genomes of the *P. damicornis* BMCs used by Rosado and colleagues (7), aiming to identify potential molecular mechanisms of interaction between members of the consortium and the host, which may in turn guide further selections of novel BMCs. In this regard, our in silico analysis can support the development of a framework for the selection of customized consortia with specific BMC characteristics for specific hosts and stress conditions, which can accelerate and optimize the selection of BMC consortia. Yet, it is important to emphasize that the beneficial characteristics for corals analyzed in this work are mainly theoretical and based on the literature, which require validation by, for example, combined physiological and molecular (such as metatranscriptomics and other omics) monitoring during laboratory experiments and field trials."

Line 574- 577 unclear if the changes in gene expression for the species other than *S. radians* and *O. faveolata* were also from bacterial origin. Please clarify.

Line 576 - *Mussismilia*

Line 582 - Unclear what exactly "this differential expression" refers to; is it references 10 and 119?

Reviewer #2 (Comments for the Author):

The authors did an excellent job responding to the reviewer's comments throughout the manuscript. It was great to see an open and expanded discussion of potential caveats of the study and I agree on their framing of this work as a potential avenue for identifying putative beneficial microbes for future experimental and, importantly, transcriptomic studies. I also appreciate the time and careful consideration the authors took to address all comments thoroughly.

I have only a few additional comments for consideration, some of which may be picked up in the proof-reading stage as well. See below:

In response to my comment originally intended for lines 446-451 (original manuscript), the authors discuss in the response to reviewer's document that one of the aims of using BMCs is to increase the number of organisms that can contribute certain functions (i.e., functional redundancy) in corals undergoing stress. It would be nice to see a short mention of this in the revised manuscript text as well, perhaps most usefully in the introduction when the ideas and aims of coral probiotics/microbial manipulation are first introduced.

Line 69: "shown the concept that" should be replaced with "showed that"

Line 193: "BMCS" should be "BMCs"

Line 356: Check the subscript notation here for H₂O₂ - it may be worth a quick check that all other chemical formulas have the correct notation throughout the manuscript.

Line 362-370: In response to my original comment regarding cobalamin (original manuscript lines 434-437), the authors discuss in the response to reviewer's document the potential importance of cobalamin. While it has already been pointed out in the main text that cobalamin synthesis is beneficial in maintaining homeostasis under normal conditions (as reviewed in the associated citation 55), the authors leave it up to the reader to make the jump to how that would be important for corals undergoing stress. It would be useful to see a half or full sentence that completes the thought here to better contextualize this discussion point in the

manuscript.

Line 437: Remove the extra space between "esterase" and "hydrolyzes"

Response to reviewer and editor comments

Exploring the **potential** molecular mechanisms of interactions between a probiotic consortium and its coral host

Editor's Comments to Authors:

Dear Prof. Raquel Peixoto:

Thank you for submitting your manuscript to mSystems. We have completed our review and I am pleased to inform you that, in principle, we expect to accept it for publication in mSystems. However, acceptance will not be final until you have adequately addressed the reviewer comments.

Dear Prof. Thulani Makhwanyane,

Thank you and the reviewers for another round of positive and constructive feedback. It has been a great experience to have our manuscript evaluated by the three of you and to have had the chance to add such valuable contributions.

Please see below our point-by-point responses to the reviewers' comments.

Reviewer comments:

Reviewer #1 (Comments for the Author):

Overall, nice changes. The goal and limitations of the study are much clearer now.

A: Thank you, we really appreciate your feedback and outstanding contribution.

Some minor points on the revisions:

Lines 35-37 it's still a little unclear how this relates to the previous sentences. I would just cut it or streamline it and make it part of the previous sentence.

A: Thank you for pointing this out. We did streamline it, and the sentence now reads as follows:

“Corals are model examples because associated microorganisms play a critical role in their development and growth (2, 3), in the control of pathogens (4-7), and in biogeochemical cycles (8) such as nitrogen, carbon, and sulfur cycling (9, 10), **also providing access to different types of nutrients (11).**”

Lines 84-90: While I'm happy to see this inclusion, this reads slightly awkwardly because it goes back and forth between limitations and the goal of the current paper. It is also unclear what exactly the last sentence adds that's not already included earlier in the paragraph. Consider trimming to something like this:

"The purpose of this work is to perform an in silico analysis of the genomes of the *P. damicornis* BMCs used by Rosado and colleagues (7), aiming to identify potential molecular mechanisms of interaction between members of the consortium and the host, which may in turn guide further selections of novel BMCs. In this regard, our in silico analysis can support the development of a framework for the selection of customized consortia with specific BMC characteristics for specific hosts and stress conditions, which can accelerate and optimize the selection of BMC consortia. Yet, it is important to emphasize that the beneficial characteristics for corals analyzed in this work are mainly theoretical and based on the literature, which require validation by, for example, combined physiological and molecular (such as metatranscriptomics and other omics) monitoring during laboratory experiments and field trials."

A: This paragraph has been edited, and it now reads exactly as suggested.

Line 574- 577 unclear if the changes in gene expression for the species other than *S. radians* and *O. faveolata* were also from bacterial origin. Please clarify.

A: Good catch, it was actually true for all of these examples, so we have edited accordingly:

"The results reveal that several genes previously suggested to be potentially beneficial to corals (and also detected in our genomes) were upregulated during stress, such as peroxidase genes related to protection against ROS (e.g., *S. radians* and *O. faveolata*), vitamin B biosynthesis (e.g., *P. clivosa*) (119), and host immune response elicited by shifts in the microbiome (10). Genes related to sulfur and nitrogen metabolism were also upregulated, with a higher expression in *S. Radians*. This differential expression was mainly due to shifts in the microbiome, suggesting that the microbiome is primarily responsible for the upregulation of these genes in the studied corals (10, 119)."

Line 576 – *Mussismilia*

A: Fixed.

Line 582 - Unclear what exactly "this differential expression" refers to; is it references 10 and 119?

A: Yes, we have included the references to clarify it, as follows:

"The differential expression was mainly due to shifts in the microbiome, suggesting that the microbiome is primarily responsible for the upregulation of these genes in the studied corals (10, 119)."

Reviewer #2 (Comments for the Author):

The authors did an excellent job responding to the reviewer's comments throughout the manuscript. It was great to see an open and expanded discussion of potential caveats of the study and I agree on their framing of this work as a potential avenue for

identifying putative beneficial microbes for future experimental and, importantly, transcriptomic studies. I also appreciate the time and careful consideration the authors took to address all comments thoroughly.

A: Thank you, we are glad to read it. Thank you for the invaluable contribution.

I have only a few additional comments for consideration, some of which may be picked up in the proof-reading stage as well. See below:

In response to my comment originally intended for lines 446-451 (original manuscript), the authors discuss in the response to reviewer's document that one of the aims of using BMCs is to increase the number of organisms that can contribute certain functions (i.e., functional redundancy) in corals undergoing stress. It would be nice to see a short mention of this in the revised manuscript text as well, perhaps most usefully in the introduction when the ideas and aims of coral probiotics/microbial manipulation are first introduced.

A: We agree and included the following sentence: "Different members of the coral microbiome may perform the same beneficial functions (i.e., functional redundancy) (10). Therefore, one of the aims of using BMCs is to increase (or retain, as some beneficial microbes may be replaced by pathogens, during thermal stress events) the number of common, abundant and native microorganisms that can contribute certain functions in corals undergoing stress (10, 26)."

Line 69: "shown the concept that" should be replaced with "showed that"

A: Fixed.

Line 193: "BMCS" should be "BMCs"

A: Fixed.

Line 356: Check the subscript notation here for H₂O₂ - it may be worth a quick check that all other chemical formulas have the correct notation throughout the manuscript.

A: Good catch, this typo has been fixed and we have also double checked the entire manuscript.

Line 362-370: In response to my original comment regarding cobalamin (original manuscript lines 434-437), the authors discuss in the response to reviewer's document the potential importance of cobalamin. While it has already been pointed out in the main text that cobalamin synthesis is beneficial in maintaining homeostasis under normal conditions (as reviewed in the associated citation 55), the authors leave it up to the reader to make the jump to how that would be important for corals undergoing stress. It

would be useful to see a half or full sentence that completes the thought here to better contextualize this discussion point in the manuscript.

A: Very good suggestion. We have slightly edited this paragraph to complete the thought, as follows:

“Vitamin B12 biosynthesis is associated with several metabolic pathways, including the generation of glutathione and DMSP antioxidants (77), which are mechanisms that can directly help corals during heat stress by mitigating increased concentrations of ROS (26, 55), and has therefore been recently suggested as a BMC trait (55). In addition, Symbiodiniaceae is not capable of synthesizing cobalamin, so adding BMCs that may play this role would help to meet the needs of endosymbiotic algae.”

Line 437: Remove the extra space between "esterase" and "hydrolyzes"

A: This extra space was deleted and we have also double checked the manuscript for any other typos and extra space.

January 4, 2023

Prof. Raquel Peixoto
King Abdullah University of Science and Technology
Thuwal
Saudi Arabia

Re: mSystems00921-22R2 (Exploring the potential molecular mechanisms of interactions between a probiotic consortium and its coral host)

Dear Prof. Raquel Peixoto:

Your manuscript has been accepted, and I am forwarding it to the ASM Journals Department for publication. For your reference, ASM Journals' address is given below. Before it can be scheduled for publication, your manuscript will be checked by the mSystems production staff to make sure that all elements meet the technical requirements for publication. They will contact you if anything needs to be revised before copyediting and production can begin. Otherwise, you will be notified when your proofs are ready to be viewed.

If you would like to submit a potential Featured Image, please email a file and a short legend to mSystems@asmusa.org. Please note that we can only consider images that (i) the authors created or own and (ii) have not been previously published. By submitting, you agree that the image can be used under the same terms as the published article. File requirements: square dimensions (4" x 4"), 300 dpi resolution, RGB colorspace, TIF file format.

We recognize that the video files can become quite large, and so to avoid quality loss ASM suggests sending the video file via <https://www.wetransfer.com/>. When you have a final version of the video and the still ready to share, please send it to mSystems staff at mSystems@asmusa.org.

Sincerely,

Thulani Makhalanyane
Editor, mSystems

Journals Department
E-mail: mSystems@asmusa.org